# Observation of perfect absorption in hyperfine levels of molecular spins with hermitian subspaces

Claudio Bonizzoni [1,2] ✉, Daniele Lamberto [3], Samuel Napoli [3], Simon Günzler [4], Dennis Rieger [4], Fabio Santanni [5], Alberto Ghirri [2], Wolfgang Wernsdorfer [4], Salvatore Savasta [3] & Marco Affronte [1,2]

We investigate Perfect Absorption (PA) of radiation, in which incoming energy is entirely dissipated, in a system consisting of molecular spin centers coherently coupled to a planar microwave resonator operated at milliKelvin temperature and in the single photon regime. This platform allows us to fine tune the spin-photon coupling and to control the effective dissipation of the two subsystems towards the environment, thus giving us the opportunity to span over a wide space of parameters. Our system can be effectively described by a non-Hermitian Hamiltonian exhibiting distinct Hermitian subspaces. We experimentally show that these subspaces, linked to the presence of PA, can be engineered through the resonator-spin detuning, which controls the composition of the polaritons in terms of photon and spin content. In such a way, the required balance between the feeding and the loss rates is effectively recovered even in the absence of PT-symmetry. We show that Hermitian subspaces influence the overall aspect of coherent spectra of cavity QED systems and enlarge the possibility to explore non-Hermitian effects in open quantum systems. We finally discuss how our results can be potentially exploited for applications, in particular as single-photon switches and modulators.

Passive open light-matter quantum systems offer the opportunity to investigate non-Hermitian physics[1–4]. Non-Hermiticity provides an insightful theoretical framework to understand unconventional phenomena such as wave-scattering anomalies, e.g., coherent perfect absorption (CPA)[5–7], perfect absorption (PA, often also referred to as Reflectionless Scattering Modes, RSM, in single-port configuration)[8–10], and electromagnetically-induced transparency[11,12], which have attracted a great deal of interest in the last decades for their potential applications in photonics [13,14]. These peculiar wave scattering effects are often associated with the presence of Exceptional Points (EP, i.e., degeneracy of the complex resonance frequencies of the system)[7,9,15], Parity-time (PT) symmetry[13,16–19], anti-PT-symmetry[20–23], and Bound

States in the Continuum (BIC) [23–25]. It is known that coherent scattering of electromagnetic radiation is fully described by the positions of the poles and zeros of the scattering matrix[6,8,25–27]. In particular, for the reflection element of the scattering matrix (e.g., $S_{11}$), the relevant spectral features are determined by the eigenvalues of two closely related, non-Hermitian effective Hamiltonians[8,25,27]. Following the convention adopted in ref. 27, we will denote these as $\hat{H}_{res}$ and $\hat{H}_{RZ}$, associated with the poles and zeros of the reflection coefficient, respectively (see Section "Theoretical modeling" and Section "Methods"). PA can occur only when one or more eigenvalues of $\hat{H}_{RZ}$ lie on the real axis, and the frequency $\omega$ of the input field matches such real eigenvalues. In ideal passive PT-symmetric systems (with respect to

[1]Department of Physics, Informatics and Mathematics, University of Modena and Reggio Emilia, Modena, Italy. [2]Institute of Nanoscience (NANO), National Research Council (CNR), Modena, Italy. [3]Department of Mathematics and Computer Sciences, Physical Sciences and Earth Sciences, University of Messina, Messina, Italy. [4]Institute of Physics, Karlsruhe Institute of Technology, Karlsruhe, Germany. [5]Department of Chemistry Ugo Schiff, University of Florence, Sesto Fiorentino (FI), Italy. ✉e-mail: claudio.bonizzoni@unimore.it

$\hat{H}_{RZ}$), where the losses and effective gains introduced through the input port are balanced, both the two relevant eigenvalues of $\hat{H}_{RZ}$ are real in the symmetry-unbroken phase, thus enabling the realization of PA or CPA. The full absorption of an input signal, which is often regarded as an undesired effect, can actually improve the performance of operations, such as detection[28,29], quantum state transfer and wavelength conversion. In this way, it may offer new opportunities in the development of optical modulators and switches or quantum sensing.

PA or CPA have been observed in optics using microdisks[7], slab waveguides[30], optical metamaterials[31,32], slabs of conductive materials[33], or nanostructured semiconductors[6]. These systems are constituted by passive components and are fed with one or more input signals. The necessary balance between incoming energy and losses is achieved by adjusting the coupling of the system with its feeding lines and/or the losses of the system. PA can also be realized at microwave frequencies using metamaterials[34], conducting films[35,36], microwave resonators[1,2], or dielectrics[37]. Similar results have been obtained using microwave cavities coupled to ferromagnetic Yttrium-Iron-Garnet (YIG) spheres[4,12,38]. Here, the system typically has a two-port configuration, which is used to send microwave excitations and, due to the low damping of YIG, its magnetic coupling with the cavity is adjusted to obtain the necessary critical coupling condition.

Diluted spin centers in a non-magnetic matrix offer the possibility to play with multiple degrees of freedom, for instance, by choosing the crystal field or exploiting nuclear spins. These features can be tailored at the synthetic level in molecular spin systems[39–42], leading to some extensive control of their coherence times and over genuine quantum features[43–45], which, in turn, translates into large potential for

applications in quantum computing[46–49] and in quantum sensing[50–52]. Molecular spins and, more in general, spin qubits in solids inevitably operate in a dissipative environment and require a proper tailoring of the photon-spin qubit transduction. Thus, having a deep control of the key experimental parameters and a full description of the theoretical problem is highly desirable to design quantum computing or quantum sensing experiments. For instance, finding optimal conditions for qubit encoding typically requires tailoring the environmental bath/s[53] or being insensitive to it and/or its fluctuations, as for *atomic clock transitions*[54]. Conversely, running molecular spins as quantum sensors requires finding conditions for which small perturbations of the external parameters lead to large signal variations, e.g., those obtained at singularities in the scattering parameters. Finally, since molecular spins can be embedded into hybrid spin-superconducting circuits[55–58], they offer an ideal testbed to investigate open non-Hermitian systems at microwave frequency.

In this work, we theoretically and experimentally investigate an open passive system that is not PT-symmetric. Our geometry consists of a planar superconducting microwave resonator in the purely quantum regime (milliKelvin temperature and average single microwave photon number) overcoupled to its input-output line and magnetically coupled to molecular spin centers, as depicted in Fig. 1. We first consider a diluted α, γ-bisdiphenylene-β-phenylallyl (BDPA, for short) organic radical spin as a prototypical Two-level-System (TLS) and, then, a tetraphenylporphyrinato oxovanadium(IV) complex (VOTPP, for short), with large ($I = 7/2$) nuclear spin and anisotropic hyperfine tensor to enlarge the number of subsystems investigated (Fig. 1d). Tuning the position of the sample on the resonator allows us to change the coupling rate with the spins, $g_\mu$, thus crossing from the strong to the weak coupling regime.

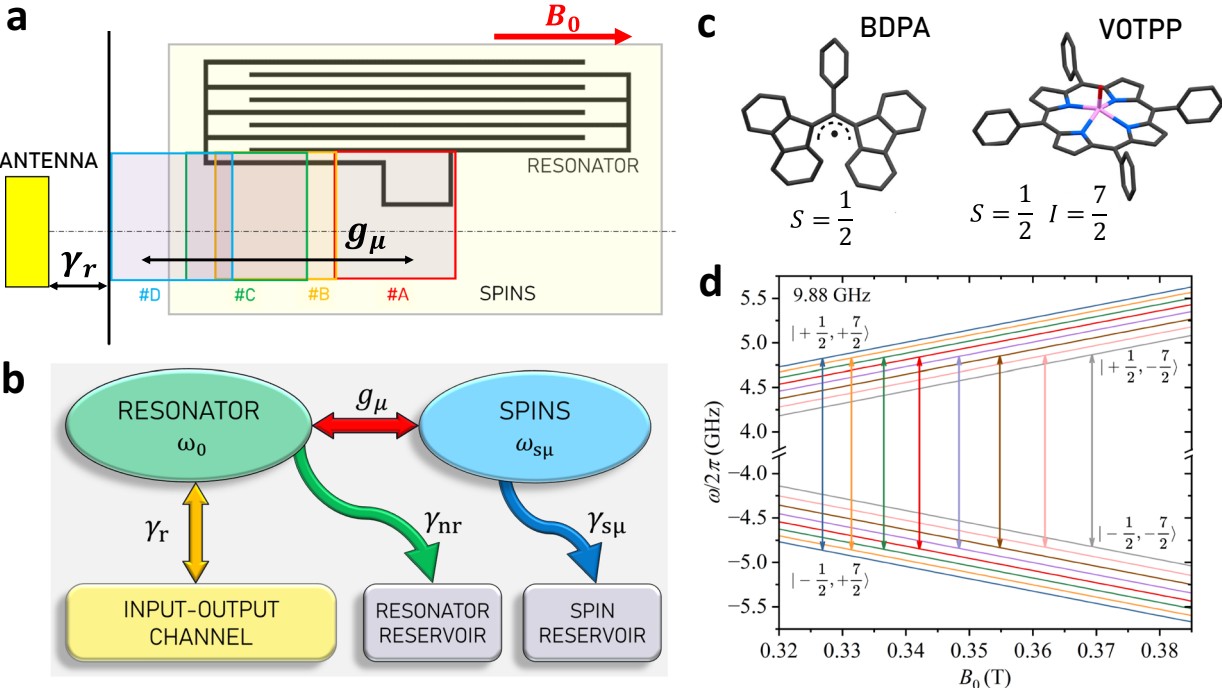

**Fig. 1 | Implementation and modeling of the open passive quantum system. a** Sketch of the lumped element resonator with all sample positions investigated in this work. The large light-yellow rectangle represents the BDPA sample, while the smaller rectangles represent the different positions of the VOTPP crystal (#A to #D, from red to light-blue). The distance between the antenna (yellow) and the chip can be adjusted at room temperature to vary the radiative relaxation rate of the resonator $\gamma_r$. The position of the sample controls the coupling strength $g_\mu$ to the resonator. The red arrow shows the direction of the applied static magnetic field, $B_0$. **b** Model adopted in

this work for the open quantum system in (**a**). The resonator is coupled to both spins and its input/output line (antenna). Only a single spin $\mu$-th ensemble is shown for clarity. **c** Molecular structure for BDPA and VOTPP. Labels indicate their electronic $S = 1/2$ and nuclear $I = 7/2$ spins. Molecular structures reproduced from refs. 40,76. **d** Easyspin simulation of the $\{S_z, I_z\}$ energy levels of VOTPP obtained with the parameters and the Hamiltonian reported in refs. 40,56 at 9.88 GHz. Vertical arrows show the eight allowed $|-\frac{1}{2}, I_z\rangle \leftrightarrow |\frac{1}{2}, I_z\rangle$ transitions giving the $\mu$-th (sub)ensembles, denoted with different colors. The vertical energy scale is cut for clarity.

Moreover, the resulting non-equal thermal population of the hyperfine multiplet of VOTPP allows us to map the effect of slightly different couplings by simply tuning the applied static magnetic field. We experimentally demonstrate that it is possible to obtain PA with zero reflection dips at energy values detuned from resonance, at two symmetric positions with respect to it. On the one hand, this phenomenology provides a direct experimental confirmation consistent with theoretical predictions in ref. 26. On the other hand, our detailed theoretical model and analysis allow us to directly link it to non-Hermitian physics. More specifically, we find that PA corresponds to tailor effective Hermitian subspaces in the non-Hermitian Hamiltonian $\hat{H}_{RZ}$, even when radiative and non-radiative losses are not balanced, thus in the absence of PT-symmetry. This explicitly links PA with the concept of Hermitian subspaces and, at the same time, the presence of these subspaces largely shapes the overall aspect of the coherent spectra of light-matter systems as a function of detuning.

## Results

### Theoretical modeling

We begin by presenting the model developed for our system, which is sketched in Fig. 1a, b) (see Section "Methods" for details). The resonator has frequency $\omega_0$, intrinsic (non-radiative) losses, $\gamma_{nr}$, and it is coupled to an antenna acting as an input-output port with rate $\gamma_r$. The $\mu$-th spin ensemble is coupled to the resonator through its collective coupling strength $g_\mu$, and has energy $\omega_{s\mu} = g_{L\mu}\mu_B B_0/\hbar$ (being $B_0$ the applied static magnetic field, $g_L$ is an effective Landé g-factor and $\mu_B$ is the Bohr's magneton) and intrinsic relaxation rate $\gamma_{s\mu}$. While BDPA can be effectively described using a single resonance frequency ($\mu = 1$), the VOTPP sample comprises multiple molecular spin levels due to hyperfine splitting ($\mu = 1, ..., 8$), in Fig. 1d. These multiple-spin ensembles can be effectively described using the Holstein-Primakoff mapping with $N$ non-interacting bosons. In the low-excitation regime, such mapping can be linearized, leading to the harmonic Hamiltonian

$$\hat{H}_S = \hbar\omega_0\, \hat{a}^\dagger \hat{a} + \hbar \sum_{\mu=1}^{N} \omega_{s\mu} \hat{b}_\mu^\dagger \hat{b}_\mu + \hbar \sum_{\mu=1}^{N} g_\mu(\hat{a}^\dagger \hat{b}_\mu + \hat{b}_\mu^\dagger \hat{a})\,, \quad (1)$$

where $\hat{a}$ and $\hat{b}_\mu$ denote the bosonic annihilation operators of the resonator and the $\mu$-th spin ensemble, respectively. The coupling strengths $g_\mu$ observed are sufficiently small with respect to $\omega_0$ to justify the use of the Rotating Wave Approximation[59]. Equation (1) can describe either BDPA for $N = 1$ or VOTPP for $N = 8$. The coupling with the external environment, which consists of the decay channels of the resonator (radiative and non-radiative) and of the spin ensembles (Fig. 1b), can be conveniently described by Heisenberg–Langevin equations, which take into account also the coherent feeding field through the antenna (see Supplementary Information). The complex reflection scattering parameter, $S_{11}(\omega)$, can be shown to have poles (i.e., resonance states) corresponding to the complex eigenfrequencies, $\Omega_j$, of the non-Hermitian Hamiltonian $\hat{H}_{res}/\hbar = \hat{\alpha}^\dagger(\mathbf{A} - i\mathbf{\Gamma}/2)\hat{\alpha}$, where $\hat{\alpha}^T = (\hat{a}, \hat{b})$, $\mathbf{A}$ is the Hopfield matrix of Hamiltonian in Eq. (1) and $\mathbf{\Gamma}$ is the corresponding decay matrix. Moreover, we observe that the zeros of $S_{11}(\omega)$ exhibit the analogous structure as the denominator, with the only difference being the reversal of the sign of the decay rate associated with the input-output port, $\gamma_r$ (see Section "Methods"[8,26,27,60]). Therefore, these reflection zeros can be interpreted as the complex eigenfrequencies, $\tilde{\Omega}_j$, of the effective non-Hermitian Hamiltonian

$$\hat{H}_{RZ}/\hbar = \hat{\alpha}^\dagger(\mathbf{A} - i\tilde{\mathbf{\Gamma}}/2)\hat{\alpha}\,, \quad (2)$$

where the decay matrix $\tilde{\mathbf{\Gamma}}$ differs from $\mathbf{\Gamma}$ only for the inversion of the sign of the input-output port decay rate (notice that $-\gamma_r$ corresponds

to the feeding rate). Hence, the reflection scattering parameter reads:

$$S_{11}(\omega) = \frac{\det\left(\hat{H}_{RZ} - \omega\hat{I}\right)}{\det\left(\hat{H}_{res} - \omega\hat{I}\right)} = \prod_{j=1}^{N+1} \frac{\omega - \tilde{\Omega}_j}{\omega - \Omega_j}\,, \quad (3)$$

where $\hat{I}$ is the identity operator. Here we notice that, in Eq. (3), we have implicitly performed an analytic continuation of the reflection scattering parameter $S_{11}(\omega)$ into the complex $\omega$-plane. However, in the context of our experimental implementation, $\omega$ must be real-valued.

Equation (3) explicitly highlights the connection between the PA condition ($S_{11}(\omega) = 0$) and the eigenvalues of the effective non-Hermitian Hamiltonian $\hat{H}_{RZ}$ in Eq. (2). Indeed, when the imaginary part of one of the complex eigenfrequencies of Eq. (2) vanishes ($\Im(\tilde{\Omega}_j) = 0$ for some $j$), it is always possible to nullify the numerator of $S_{11}$ by appropriately tuning the frequency of the input probing field $\omega$. Reflection zeros on the real axis are also referred to as Reflectionless Scattering Modes (RSM)[8,27]. To further elucidate this point, we first perform a rotation on the bare bosonic operators to transform them in the polariton basis (normal modes). Specifically, we introduce a suitable Hopfield transformation $\mathbf{U}$, defined as $\hat{\mathbf{P}} = \mathbf{U}\hat{\alpha}$, such that the Hamiltonian of the isolated system in Eq. (1) takes the diagonal form $\hat{H}_S = \sum_{j=1}^{N+1} \bar{\omega}_j \hat{P}_j^\dagger \hat{P}_j$. Subsequently, the effective non-Hermitian Hamiltonian describing the open system dynamics of the reflection zeros, in the strong coupling regime, is given by

$$\hat{H}_{RZ} = \sum_{j=1}^{N+1} \left(\bar{\omega}_j - i\frac{\bar{\gamma}_j}{2}\right)\hat{P}_j^\dagger \hat{P}_j\,, \quad (4)$$

where the effective (or *dressed*) loss rates $\bar{\gamma}_j$ are defined by the diagonal elements of $\mathbf{U}\tilde{\mathbf{\Gamma}}\mathbf{U}^\dagger$ (see Section "Methods"). Equation (4) provides a clear physical interpretation of the reflection zeros in terms of the polariton modes. In particular, the PA condition turns out to be $\bar{\gamma}_j = 0$ for some $j$. We remark that this effective Hamiltonian is derived under the hypothesis of PA and is valid in the strong coupling regime, as, in this regime, the mixed terms in the decay channels can be safely neglected due to the different resonance frequencies of the polariton peaks.

### The single ensemble case

We begin by analyzing the BDPA organic radical. Figure 2a displays experimental reflection spectra maps as a function of the static magnetic field. Figure 2b illustrates the corresponding theoretical fit performed using Eq. (3) with $N = 1$, showing excellent agreement. An anticrossing, which is typical of the strong spin–photon coupling regime, is clearly visible when $\omega_s$ crosses $\omega_0$. Fits gives $g/2\pi \approx 20.7$ MHz and $\gamma_s/2\pi = 5$ MHz, further corroborating the strong coupling regime (see Supplementary Table 1 for full fit parameters). We observe the presence of two dips (Fig. 2c), approaching zero reflection, corresponding to PA. Remarkably, these occur before (0.3443T) and after (0.3459T) the resonant magnetic field value, corresponding to symmetric resonator-spin detuning $\Delta = \omega_s - \omega_0 \approx \pm 8 \cdot 10^{-4}$ T. This clearly differs from what is typically observed in systems where the corresponding $\hat{H}_{RZ}$ is PT-symmetric, in which the dips occur at resonance[4,7]. The positions and the absolute values of the two reflection zeros are also predicted by our model (Fig. 2b). As a first important result, this demonstrates (both experimentally and theoretically) that PA can be realized on molecular spin centers by tuning their resonance frequency. More specifically, in the strong coupling regime, the imaginary part of the $j$th effective complex eigenfrequencies, $\Im(\tilde{\Omega}_j)$, is approximately equal to the effective loss rates $\bar{\gamma}_j/2$, as given by Eq. (4), which, in turn, can be expressed by a linear combination of the single-channel loss rates (see Section "Methods"):

$$\bar{\gamma}_j = (-\gamma_r + \gamma_{nr})|U_{j1}|^2 + \gamma_s|U_{j2}|^2 = -\bar{\gamma}_{cj} + \bar{\gamma}_{sj}\,. \quad (5)$$

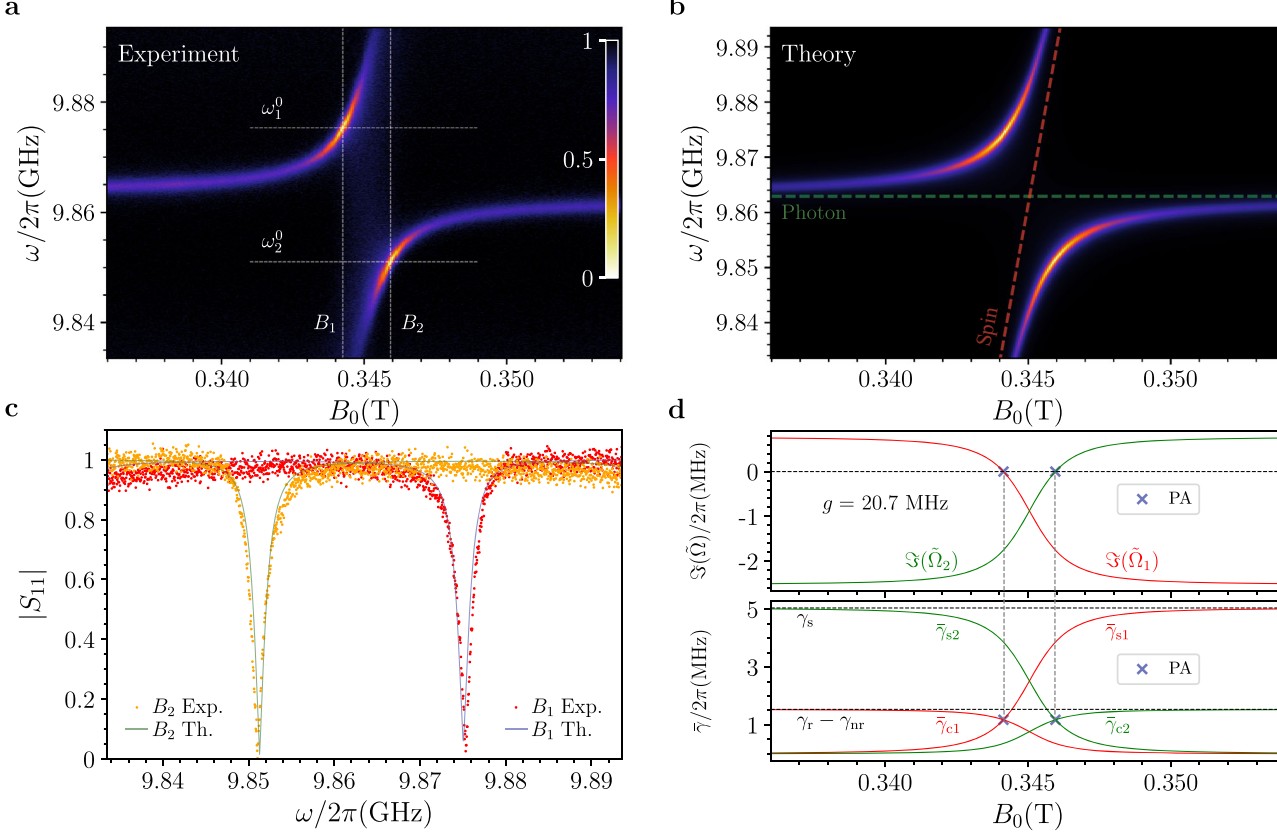

**Fig. 2 | Perfect Absorption for the single ensemble case. a** Normalized reflection map ($|S_{11}(\omega)|$) measured for the BDPA sample at 25 mK as a function of the static magnetic field, $B_0$. **b** Simulated reflection map obtained using the fit parameters extracted from the map in (**a**), according to Eq. (8) in Section "Methods" (fit parameters are reported in Supplementary Table 1). **c** Theoretical (green and blue lines) and experimental (orange and red dots) normalized reflection spectra extracted from the maps in **a**, **b**, showing two dips with nearly zero reflection (see vertical lines $B_1$ and $B_2$ in (**a**)). **d** (Upper panel) Imaginary parts of $\tilde{\Omega}_{1,2}$, $[\Im(\tilde{\Omega}_{1,2})]$, as a function of $B_0$, calculated using Eq. (3) and the fit parameters obtained from (**a**). Perfect Absorption (blue crosses) occurs at the polariton resonances when $\Im(\tilde{\Omega}_{1,2})$ the crosses zero. **d** Dressed cavity feeding rate $\bar{\gamma}_{ci}$ and dressed spin loss rate $\bar{\gamma}_{si}$ as functions of $B_0$, computed according to Eq. (5). Perfect Absorption (blue crosses) is achieved when $\bar{\gamma}_{ci} = \bar{\gamma}_{si}$ for the $i$th polariton (see main text).

Here, $U_{jk}$ are the Hopfield coefficients defined by **U**, which determines the degree of photon ($|U_{j1}|^2$) and spin ($|U_{j2}|^2$) hybridization of the polaritons, and which can be tuned through the spin-resonator detuning. $\bar{\gamma}_{cj} = (\gamma_r - \gamma_{nr})|U_{j1}|^2$ and $\bar{\gamma}_{sj} = \gamma_s |U_{j2}|^2$ represent the *dressed cavity* feeding rate and the *dressed spin* loss rate, respectively. In our experiments, we vary the detuning, $\Delta$, through the externally applied magnetic field. Therefore, PA (defined by $\bar{\gamma}_j = 0$) occurs when the balance between *dressed feeding and loss* rates is achieved, analogously to the balance condition in PT-symmetric systems for the *bare* rates, even if the $\hat{H}_{RZ}$ Hamiltonian does not display such symmetry (see Section "Discussion" for a more detailed comparison with PT-symmetric systems). Equation (5) can be satisfied for either the lower or upper polariton, leading to the emergence of a Hermitian subspace within the Hilbert space of $\hat{H}_{RZ}$. This subspace $\mathcal{H}^{(j)}$ is effectively described by the Hamiltonian $\hat{H}_{HS}^{(j)} = \bar{\omega}_j \hat{P}_j^\dagger \hat{P}_j$, where $j = 1$ or $j = 2$ for the upper or lower polariton, respectively. Furthermore, it can be easily shown that if a Hermitian subspace exists for one of the polariton modes at a given detuning $\Delta'$, a second one will exist at $-\Delta'$ for the other polariton mode. This property holds even in the case of multiple spin resonances, as it will be shown for the VOTPP sample. Figure 2d shows the imaginary part of the complex eigenvalues of Eq. (2), $\Im(\tilde{\Omega}_j)$, along with the corresponding contributions to the $j$th polariton mode of the dressed cavity feeding and spin loss rates. The balance between the two contributions ($\bar{\gamma}_{cj} = \bar{\gamma}_{sj}$), corresponding to the simultaneous vanishing of $\Im(\tilde{\Omega}_j)$, is observed for either of the polariton modes at symmetric detunings, as discussed above.

## Effects of coupling strength

We now investigate theoretically the effect of the coupling strength on PA. Figure 3 shows reflection maps simulated with Eq. (8) by using the relaxation rates obtained from the fits of Fig. 2, along with different coupling strengths. Notably, as the coupling decreases, the dips progressively merge towards the resonant field value. We identify a threshold value, $g_{th} = (\gamma_r - \gamma_{nr} + \gamma_s)/4$ (green line in Fig. 3c), marking the transition from the strong to the weak coupling regime. An intermediate region emerges, where no crossing of the imaginary parts occurs anymore but PA still persists (yellow line), until the coupling strength reaches a minimum value $g_{min} = \sqrt{(\gamma_r - \gamma_{nr})\gamma_s}/2$, at which $\Im(\tilde{\Omega}_j)$ exhibits a double root at zero detuning (blue line) and the reflection dips coalesce into a single one. For couplings below $g_{min}$, the imaginary part $\Im(\tilde{\Omega})$ cannot cross the zero, preventing the observation of PA (red line). The simulations in Fig. 3 suggest that our model in Section "Theoretical modeling ", although developed under the assumption of a strong coupling regime, also holds in the weak coupling regime.

## The multiple ensemble case

We now experimentally investigate an analogous phenomenology using the VOTPP sample, which, in contrast to BDPA, exhibits

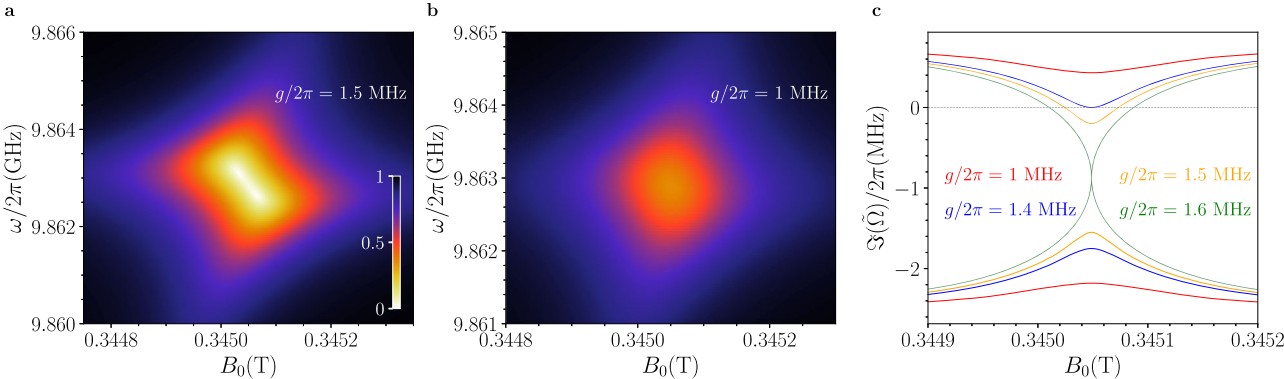

**Fig. 3 | Perfect Absorption during the transition from strong to weak coupling regime. a**, **b** Normalized reflection ($|S_{11}|$) maps as a function of the static magnetic field $B_0$, simulated using Eq. (8) in Section "Methods" with the parameters fitted from the data in Fig. 2, except for using lower coupling strength values of $g/$

$2\pi = 1.5$ MHz (**a**) and $g/2\pi = 1$ MHz (**b**), respectively. **c** Imaginary parts of $\tilde{\Omega}_{1,2}$, $[\Im(\tilde{\Omega}_{1,2})]$, calculated as a function of the static magnetic field using the relaxation rates fitted from the data in Fig. 2 and four different values of $g/2\pi$. Perfect absorption occurs when $\Im(\tilde{\Omega}_{1,2})$ the cross-zero and cannot be realized for $g/2\pi < 1.4$ MHz (see main text).

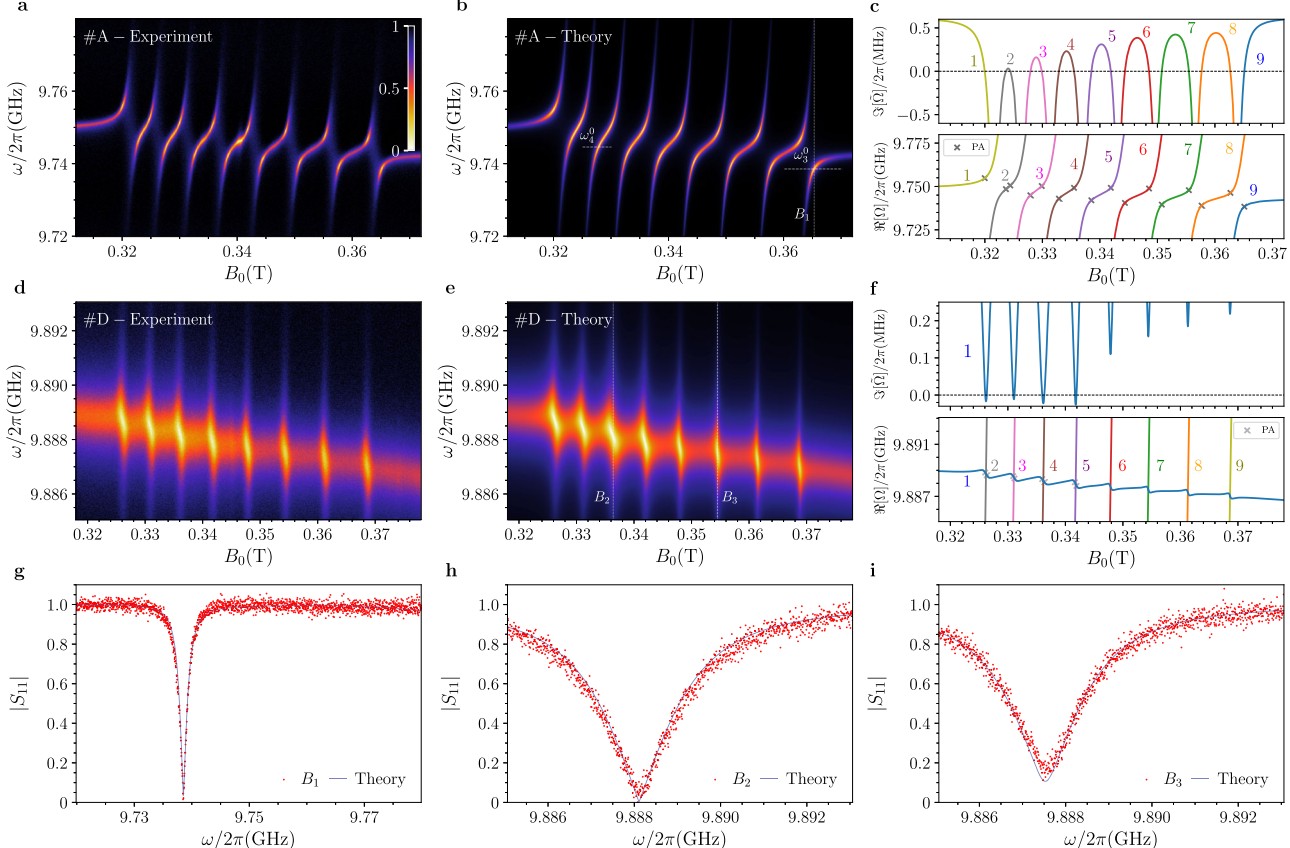

**Fig. 4 | Perfect Absorption for the multiple spin case. a**, **d** Normalized reflection ($|S_{11}|$) maps measured as a function of the static magnetic field $B_0$ at 30 mK for the VOTPP crystal. PA is observed in the proximity of multiple hyperfine levels. The strongest (position #A) and the weakest (position #D) coupling regimes are shown, respectively. **b**, **e** Simulated reflection maps obtained by fitting the maps in **a**, **d** according to Eq. (8) in Section "Methods". **c**, **f** (Upper panels) Imaginary parts of the complex frequencies, $\Im(\tilde{\Omega}_j)$, as a function of the magnetic field $B_0$, for the jth polariton frequency. The horizontal black dotted line at $\Im(\tilde{\Omega}) = 0$ corresponds to the PA condition. **c**, **f** Real parts of the polariton frequencies, $\Re(\Omega_j)$, with the predicted PA points, showing excellent agreement with the experimental data. Both $\Im(\tilde{\Omega}_j)$ and $\Re(\Omega_j)$ are calculated using (3) supported by (8) in Section "Methods". **g**–**i** Experimental (red dots) and theoretical (blue lines) normalized reflection obtained according to the vertical lines shown in **b**, **e** ($B_1$, $B_2$, and $B_3$), displaying perfect absorption dips and one not satisfying this condition. All fit parameters are given in Supplementary Tables 5 and 8, respectively.

multiple resonance frequencies, $\omega_{s\mu}$. As mentioned in the introduction, the overall coupling regime for the whole group of transitions is primarily determined by the position of the sample with respect to the resonator (see Fig. 1). In addition, the values of the hyperfine tensor of VOTPP (see Section "Methods") give different

thermal population of hyperfine levels at milliKelvin temperature, thus allowing us to span a large set of $g_j$ values within a unique field scan.

Figure 4a, d shows experimental reflection maps, $|S_{11}|$, measured as a function of the static magnetic field, for positions #A and

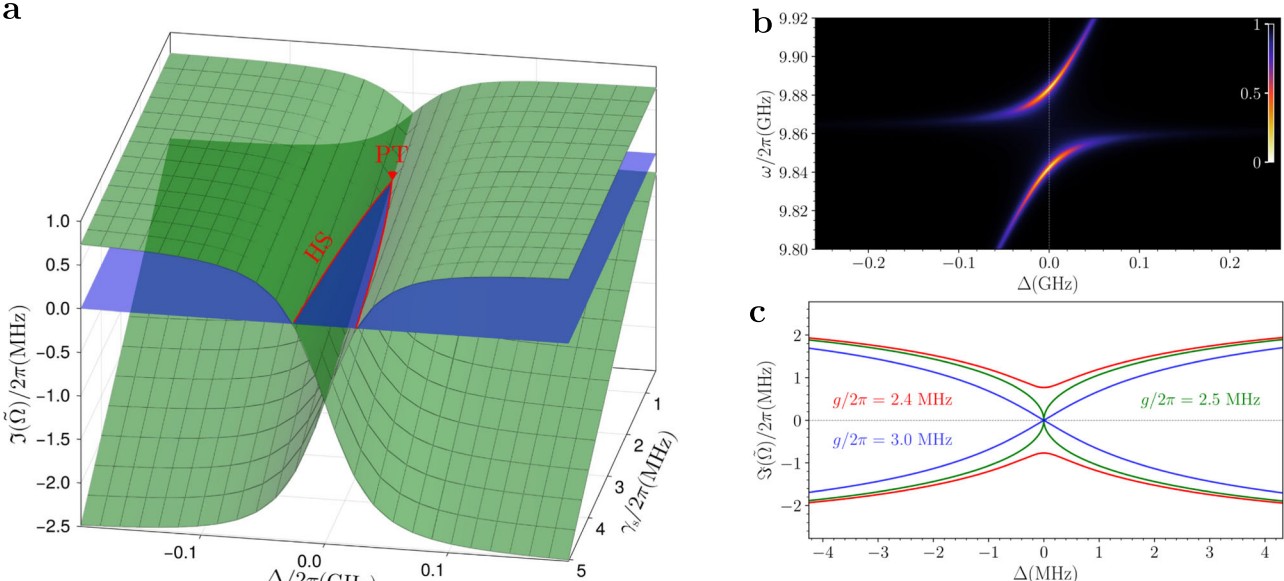

**Fig. 5 | Comparison with PT-symmetry. a** Surface plot showing the imaginary part of the complex eigenvalues of the effective non-Hermitian Hamiltonian $\hat{H}_{RZ}$ as a function of the detuning, $\Delta$, and the spins decay rate, $\gamma_s$. The simulation is obtained using $N = 1$ and the values reported in Supplementary Table 1. The two red lines indicate where an Hermitian subspace (HS) is realized, i.e., the intersection with the $\Im(\tilde{\Omega}) = 0$ plane (in blue), where one of the eigenvalues becomes real. These lines coalesce at the point $(\gamma_s, \Delta) = (\gamma_r - \gamma_{nr}, 0)$, marked by the red triangle, corresponding to the PT symmetry condition. **b** Normalized reflection map as a function of the detuning for a PT-symmetric system in the strong coupling regime. PA is achieved simultaneously for both polariton branches at $\Delta = 0$, conversely to the result presented in this work. **c** Imaginary parts of the eigenfrequencies $\Im(\tilde{\Omega})$ as a function of the detuning for different coupling strengths, ranging from the weak to the strong coupling regime, in a PT-symmetric system.

#D of VOTPP (see Fig. 1). In position #A, the observed avoided level crossings clearly indicate that the high cooperativity regime is achieved on each line. This is also supported by the fitted values $g_\mu/2\pi = 13–16$ MHz and $\gamma_{s\mu}/2\pi = 7–13$ MHz (depending on the line, see Supplementary Table 5). A pair of dips in reflection, approaching near-zero values and at non-zero symmetric detunings from resonance, is clearly visible for each line, indicating the presence of PA. Conversely, data for position #D show the weak coupling regime for all resonances and significantly lower coupling strengths (see Supplementary Table 8). Here, due to the different thermal population of each line, PA occurs only for a subset of the eight resonant lines (from left to right, the first four), and the coalescence of those dips is observed for increasing magnetic field values. The theoretical reflection maps in Fig. 4b, e, simulated through Eq. (3), are in excellent agreement with the data. For position #A, the PA points are perfectly predicted by the zeros of the imaginary parts of the corresponding effective eigenfrequencies (each line crosses the zero, see Fig. 4c), which are calculated by using the fit parameters obtained from Fig. 4a. Conversely, for position #D, the imaginary parts of the complex eigenvalues cross zero only for the first four lines. This can be explained by the minimal coupling $g_{min}$ discussed in Section "The Single Ensemble Case", since moving across the different resonances results in the coupling strengths $g_j$ becoming smaller than $g_{min}$. These latter results further show that Eq. (3) holds for any coupling regime, demonstrating that the PA condition can be achieved even in the weak coupling regime, provided that a minimum coupling rate value is overcome for a given set of relaxation rates.

## Discussion

We have experimentally realized PA into a passive open quantum system composed of molecular spin centers, both in the strong and weak coupling regimes to a planar superconducting microwave resonator. Our results show that spins and, more specifically, molecular spins, turn out to be an excellent testbed for

investigating non-Hermitian physics, thanks to the possibility of tuning different parameters and spanning over different coupling regimes.

We observe that our experimental results refer to an implementation for which the effective non-Hermitian Hamiltonian associated with the zeros of the reflection coefficient ($\hat{H}_{RZ}$) is not PT-symmetric. Here we notice that, in principle, PT-symmetry could be realized in these platforms by further tuning system parameters, e.g., by modifying the position of the antenna relative to the resonator, and properly tuning the static magnetic field. However, our implementation is less constrained than PT-symmetric configurations, as light-matter detuning can be easily adjusted to achieve PA without strictly satisfying the loss balance condition required in PT-symmetric systems. In this regard, we provided a simple - yet physically insightful - interpretation linking the imaginary part of the reflection zeros to the spin and photon content of the polaritons, through the Hopfield coefficients. These coefficients (and thus the position of the reflection zeros on the complex plane) can be dynamically tuned by varying the resonance frequency of one of the subsystems (in our work, by acting on the external magnetic field), which enables the movement of the reflection zeros on and off the real axis and the realization of a Hermitian subspace within the Hilbert space of $\hat{H}_{RZ}$. This flexibility makes our platform promising not only for exploring PA in coherently coupled systems, but also for studying a broader class of non-Hermitian phenomena in passive open quantum systems. These include dissipative couplings mediated by waveguides[7,27], as well as non-Hermitian topological effects[61,62].

Our theoretical framework is consistent with previous findings observed in PT-symmetric systems (e.g., refs. 4,7), of which it can be regarded as a generalization. More specifically, when the system is on resonance, $\omega_0 = \omega_s$ (i.e., $\Delta = 0$), and the decay rates satisfy $\gamma_r - \gamma_{nr} = \gamma_s \equiv \gamma$, the effective non-Hermitian Hamiltonian describing the reflection zeros, $\hat{H}_{RZ}$, becomes PT-symmetric (Fig. 5a). Under these conditions, the Hopfield coefficients satisfy $|U_{j1}|^2 = |U_{j2}|^2 = 1/2$,

a

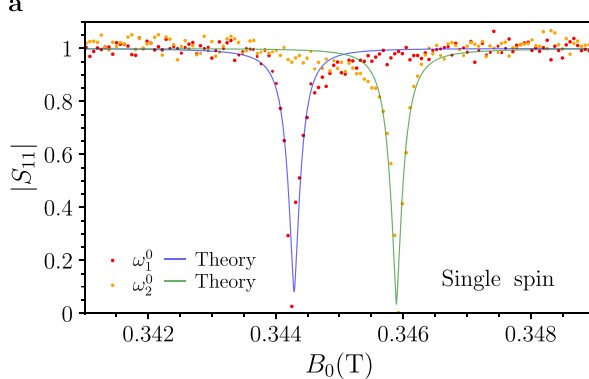

b

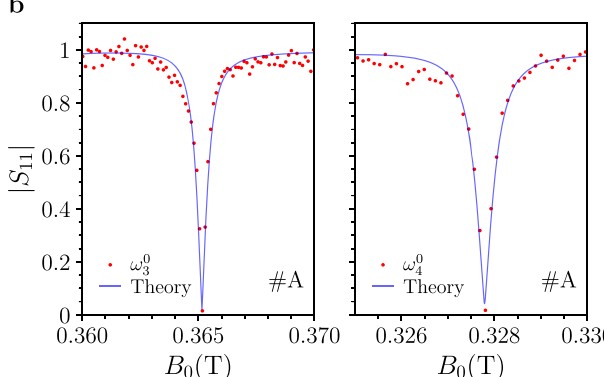

**Fig. 6 | Potential implementation of single microwave photon switches.**
**a** Experimental (red and orange dots) and theoretical (blue and green lines) normalized reflection ($|S_{11}|$) as a function of the static magnetic field $B_0$ for the BDPA sample (single spin case), highlighting Perfect Absorption. Data are extracted along the horizontal lines shown in Fig. 2a (at fixed frequency, $\omega_1^0$ and $\omega_2^0$). **b** Experimental (red dots) and theoretical (blue lines) normalized reflection, $|S_{11}|$, obtained for the VOTPP crystal (position #A) under PA and based on the horizontal lines illustrated in Fig. 4b (see $\omega_3^0$ and $\omega_4^0$).

and the balance condition in Eq. (5) is satisfied for both the polariton modes. This behavior is clearly confirmed in the reflection spectra, where both dips reach zero for $\Delta = 0$ (Fig. 5b). As the coupling strength is decreased, the system reaches the Exceptional Point at $g_{EP} = \gamma/2$ (green line in Fig. 5c). Given that in PT-symmetric systems the relation $g_{th} = g_{min} = g_{EP}$ holds, no intermediate regime appears as the coupling decreases and, thus, the presence of a crossing in the imaginary parts of the eigenvalues, $\Im(\tilde{\Omega}_i)$, can be directly associated with the presence of PA. Indeed, in PT-symmetric systems $\Im(\tilde{\Omega}_i)$ is symmetric with respect to the real axis and exhibits a double zero (at $\Delta = 0$) only for $g > g_{EP}$. Furthermore, our theoretical framework holds even in the presence of multiple resonances, as the ones of VOTPP. This goes beyond traditional PT-symmetric models, as the notion of PT-symmetry becomes ill-defined in systems with more than two coupled subsystems.

Our experiments are carried out in the purely quantum regime, which is milliKelvin temperature and average single microwave photon number in the resonator. Although PA is a general phenomenon of resonant systems, including classical ones, the model we use is perfectly suitable to describe quantum systems. For instance, this is relevant in the view of exploring how non-Hermitian physics can affect non-classical effects (see, e.g., ref. 63).

As an example for applications, we consider fast *single-photon switches or modulators* for microwave radiation[13]. Figure 6 displays reflection measurements ($|S_{11}|$) and corresponding theoretical fits, as a function of the externally applied magnetic field $B_0$ (i.e., at a fixed microwave frequency), corresponding to the horizontal line-cuts shown in Fig. 2a and Fig. 4b, e. A rather small

variation of the magnetic field ($-10^{-4}$ T) can switch the reflectivity from its maximum value to nearly zero. In this way, the reflection can be switched, suppressing the single microwave photon or allowing it to propagate by simply applying a time-dependent local magnetic field bias (e.g., through a modulation coil). We can estimate the achievable modulation depth from the data in Fig. 6 as:

$$M_d = 20 \log_{10}\left[\frac{|S_{11}^{on}|}{|S_{11}^{off}|}\right], \tag{6}$$

where $|S_{11}^{on(off)}|$ represents the maximum (minimum) value of $|S_{11}|$. Without specific optimization of the system, we find $M_d \approx 50$ dB for the BDPA by varying $B_0$ within a range of approximately $4 \cdot 10^{-4}$ T (Fig. 6a). Notably, this value remains robust against fluctuations in the *on*-state (maximum $S_{11}$), as the near-zero reflectivity in the *off*-state (minimum $S_{11}$) dominates the modulation depth. For the VOTPP sample in position #A (Fig. 6b), we observe a shallower dip compared to BDPA. However, we obtain $M_d \approx 35$ dB using lines $\mu = 3, 4$.

Further applications can be foreseen by noticing that singularities in spectra are of interest for sensing[14,64,65]. Here, a possible detection mechanism relies on the variation of the reflection signal ($|S_{11}|$) upon the strength of an added perturbation (*slope detection*[64]). For instance, based on the data in Fig. 6a, b, and without any specific optimization of the system, we can estimate a slope of $|\frac{\Delta S_{11}}{\Delta B_0}| \approx 2016 \, T^{-1}$, corresponding to a transduction coefficient of $\approx 5 \cdot 10^{-4}$ T for unit of reflection around the PA point. Here, a potential advantage of PA relies on its relatively simple phenomenology and on the reduced intrinsic noise and fluctuations, which, otherwise, could be a limiting factor for sensing[14]. Detection of electromagnetic radiation could, in principle, benefit from our results after proper extension and optimization of our system (e.g., adding an additional input transmission line or antenna to route the incoming radiation to the sensor or using photoresponsive molecular spins[39]). For instance, detection of itinerant single microwave photons would help in searching for rare events, such as *Dark Matter Axions*[65,66].

We finally mention that our results can be readily transferred and applied also on different paramagnetic spin centers, including defects such as $Er^{3+}$ ions, P donors in Si and Nitrogen-Vacancy centers, and, potentially, further extended and applied to very different frequency ranges and platforms (e.g., optical frequency).

## Methods
### Derivation of the reflection scattering parameter
To derive the reflection spectrum, we first express the output fields in terms of the input fields using the quantum Langevin equations. This leads to the general relation (see Supplementary Information)

$$\hat{\mathbf{F}}_{out}(\omega) = \left[-\frac{1}{2}\mathbf{\Gamma} - i(\omega\mathbf{I} - \mathbf{A})\right]\left[\frac{1}{2}\mathbf{\Gamma} - i(\omega\mathbf{I} - \mathbf{A})\right]^{-1}\hat{\mathbf{F}}_{in}(\omega), \tag{7}$$

where $\hat{\mathbf{F}}_{in(out)}(\omega)$ is the input (output) Langevin force vector. These vectors contain the input (output) operators associated to the different channels, i.e., the antenna (both the radiative and non-radiative components, $\hat{a}_{r,in(out)}(\omega)$ and $\hat{a}_{nr,in(out)}(\omega)$, respectively) and the spin ensembles ($\hat{b}_{\mu,in(out)}(\omega)$).

Since we focus on coherent reflection spectra where the signal enters and is sampled through the radiative port of the antenna, $S_{11}(\omega)$, we assume that $\langle\hat{a}_{nr,in}\rangle = \langle\hat{b}_{\mu,in}\rangle = 0$. Therefore, taking the expectation value of Eq. (7), where $\langle\hat{\mathbf{F}}_{in}\rangle = (\sqrt{\gamma_r}\langle\hat{a}_{r,in}\rangle, 0, \ldots)^T$, and applying the input−output relation for the radiative port,

$\hat{a}_{\mathrm{r,out}}(\omega) = \hat{a}_{\mathrm{r,in}}(\omega) - \sqrt{\gamma_{\mathrm{r}}}\hat{a}(\omega)$, we derive the reflection coefficient

$$S_{11}(\omega) = \frac{\langle \hat{a}_{\mathrm{r,out}}(\omega)\rangle}{\langle \hat{a}_{\mathrm{r,in}}(\omega)\rangle} = \frac{\frac{-\gamma_{\mathrm{r}}+\gamma_{\mathrm{nr}}}{2} - i(\omega-\omega_0) + \sum_{\mu=1}^{N}\frac{g_{\mu}^2}{\frac{\gamma_{s\mu}}{2}-i(\omega-\omega_{s\mu})}}{\frac{\gamma_{\mathrm{r}}+\gamma_{\mathrm{nr}}}{2} - i(\omega-\omega_0) + \sum_{\mu=1}^{N}\frac{g_{\mu}^2}{\frac{\gamma_{s\mu}}{2}-i(\omega-\omega_{s\mu})}}$$

$$= 1 - \frac{\gamma_{\mathrm{r}}}{\frac{\gamma_{\mathrm{r}}+\gamma_{\mathrm{nr}}}{2} - i(\omega-\omega_0) + \sum_{\mu=1}^{N}\frac{g_{\mu}^2}{\frac{\gamma_{s\mu}}{2}-i(\omega-\omega_{s\mu})}} \; . \tag{8}$$

This equation serves as a model for fitting experimental reflectivity spectra. Notably, as pointed out in Section "Theoretical modeling" and in refs. 8,27,60, the resonances of the reflection scattering coefficient $S_{11}(\omega)$ corresponds to the zeros $\Omega_j$ of the characteristic polynomial of the non-Hermitian Hamiltonian $\hat{H}_{\mathrm{res}}/\hbar = \hat{\boldsymbol{\alpha}}^{\dagger}(\mathbf{A} - i\boldsymbol{\Gamma}/2)\hat{\boldsymbol{\alpha}}$. The numerator has the same structure as the denominator, differing only in the sign of the damping rate associated with the input-output channel, $\gamma_{\mathrm{r}}$. By introducing the effective decay matrix $\bar{\boldsymbol{\Gamma}}$, which retains the structure of $\boldsymbol{\Gamma}$ but with the sign of $\gamma_{\mathrm{r}}$ reversed, the numerator can be interpreted as the characteristic polynomial of the effective non-Hermitian Hamiltonian $\hat{H}_{\mathrm{RZ}}/\hbar = \hat{\boldsymbol{\alpha}}^{\dagger}(\mathbf{A} - i\tilde{\boldsymbol{\Gamma}}/2)\hat{\boldsymbol{\alpha}}$, whose eigenfrequencies are $\tilde{\Omega}_j$.

## Derivation of the effective Hamiltonian in the strong coupling regime

In this section, we present the derivation for a single spin ensemble of the effective non-Hermitian Hamiltonian in the strong coupling regime. Starting from the quantum Langevin equations for the photon and the collective spin operators presented in the Supplementary Information, we apply the unitary transformation $\mathbf{U}$ defined in Section "Theoretical modeling" to rotate in the polariton basis, i.e., $\hat{\mathbf{P}} = \mathbf{U}\hat{\boldsymbol{\alpha}}$. Therefore, we obtain the following equations of motion for the polaritonic bosonic operators

$$\partial_t \hat{P}_j = -i\bar{\omega}_j \hat{P}_j - \int_{-\infty}^{\infty}d\omega\, U_{j1}\sum_{k=\mathrm{r,nr}}\sqrt{\frac{\gamma_k}{2\pi}}\hat{c}_k(\omega) - \int_{-\infty}^{\infty}d\omega\, U_{j2}\sqrt{\frac{\gamma_s}{2\pi}}\hat{d}(\omega),$$
$$\tag{9}$$

where $\hat{c}_k$ and $\hat{d}$ are the baths' bosonic operators, which are used to construct the corresponding input (output) operators, $\hat{a}_{k,\mathrm{in(out)}}$ and $\hat{b}_{\mathrm{in(out)}}$ (see Supplementary Information). $\bar{\omega}_j$ and $U_{jk}$ are the polaritonic eigenfrequencies and Hopfield coefficients, respectively, as defined in the main text.

Following the approach outlined in refs. 67,68, we substitute the formal solutions of the equations of motion for $\hat{c}_k(\omega)$ and $\hat{d}$ into Eq. (9). By taking the expectation value of the resulting expression, we obtain

$$\partial_t p_j(t) = -i\bar{\omega}_j p_j(t) - \frac{\gamma_j}{2}p_j(t) - U_{j1}U_{m1}^*\frac{\gamma_{\mathrm{r}}+\gamma_{\mathrm{nr}}}{2}p_m(t)$$
$$- U_{j2}U_{m1}^*\frac{\gamma_s}{2}p_m(t) + \sqrt{\gamma_{\mathrm{r}}}\,U_{j1}a_{\mathrm{in}}(t), \tag{10}$$

where $m$ is the complementary index of $j$, i.e., $m = 2$ when $j = 1$ and vice versa. In addition, we introduced the definitions $p_j(t) = \langle \hat{P}_j(t)\rangle$, $\gamma_j = (\gamma_{\mathrm{r}}+\gamma_{\mathrm{nr}})|U_{j1}|^2 + \gamma_s|U_{j2}|^2$ and $a_{\mathrm{in}}(t) = \langle \hat{a}_{\mathrm{r,in}}\rangle$, which represents the coherent input signal. Specifically, in the derivation of Eq. (10), we took into account that the system has a coherent feeding only through the antenna radiative channel, thus implying $\langle \hat{a}_{\mathrm{nr,in}}\rangle = \langle \hat{b}_{\mathrm{in}}\rangle = 0$. Eq. (10) can be rewritten in the frequency domain as

$$-i\omega p_j(\omega) = -i\bar{\omega}_j p_j(\omega) - \frac{\gamma_j}{2}p_j(\omega) - U_{j1}U_{m1}^*\frac{\gamma_{\mathrm{r}}+\gamma_{\mathrm{nr}}}{2}p_m(\omega)$$
$$- U_{j2}U_{m1}^*\frac{\gamma_s}{2}p_m(\omega) + \sqrt{\gamma_{\mathrm{r}}}\,U_{j1}a_{\mathrm{in}}(\omega). \tag{11}$$

By imposing the PA condition, $a_{\mathrm{out}}(\omega) = 0$, we obtain from the input-output relations the explicit expression $a_{\mathrm{in}}(\omega) = \sqrt{\gamma_{\mathrm{r}}}\,(U_{11}\,p_1(\omega) + U_{12}\,p_2(\omega))$, which can be inserted in Eq. (11). Furthermore, for $\omega \approx \bar{\omega}_j$, only $p_j$ is significantly excited by the input field, while the terms involving the other polariton mode can be safely neglected in the strong coupling regime, due to the great separation of the spectral lines. Hence, the effective dynamics in the time domain can be written as

$$\partial_t p_j(t) = -i\bar{\omega}_j p_j(t) - \frac{\bar{\gamma}_j}{2}p_j(t), \tag{12}$$

where $\bar{\gamma}_j = (-\gamma_{\mathrm{r}}+\gamma_{\mathrm{nr}})|U_{j1}|^2 + \gamma_s|U_{j2}|^2$, as in Eq. (5). These equations of motions can be derived by the effective non-Hermitian Hamiltonian

$$\hat{H}_{\mathrm{RZ}} = \sum_{j=1,2}\left(\bar{\omega}_j - i\frac{\bar{\gamma}_j}{2}\right)\hat{P}_j^{\dagger}\hat{P}_j, \tag{13}$$

which coincides with Eq. (4). This procedure can be easily generalized in the case of multiple spin resonances, as in the strong coupling regime, each photon-spin anti-crossing is separated from the others.

## Experimental set up

We use a planar superconducting lumped-element LC microwave resonator made of superconducting Niobium films (thickness, 50 nm) on sapphire substrate (thickness, 420 μm), as shown in Fig. 1. The resonator has a small inductive loop to enhance the generated microwave magnetic field, which is coupled to a large interdigitated capacitance. The resonator works in reflection mode, at a fundamental frequency $\omega_0/2\pi \approx 9.9$ GHz, and it is coupled to the input-output line through an antenna, whose position can be adjusted at room temperature, before cooling the sample. The chip carrying the resonator is loaded into a cylindrical copper waveguide sample holder hosting the antenna in his bottom[69,70]. Preliminary characterization of the empty resonator is reported in Supplementary Information.

All experiments are carried out inside a Qinu Sionludi dilution refrigerator (base temperature 20 mK) equipped with a three-axial superconducting magnet and microwave lines and electronics[69,70]. The input signal is attenuated by 60 dB inside the cryostat before reaching the sample box with the antenna, while the output line hosts a cryogenic High Electron Mobility Transition (HEMT) amplifier (Low Noise Factory, 37 dB gain). The signal is further amplified at room temperature before acquisition. The complex reflection scattering parameter, $S_{11}$, is measured with a Vector Network Analyzer (VNA) for different values of the static magnetic field applied, obtaining the 2D maps shown in Figs. 2 and 4. The input power at the position of the antenna is between −130 and −120 dBm, corresponding to an average single microwave photon into the resonator (see Supplementary Information). The cylindrical box and the resonator are aligned into the magnet in order to have the static magnetic field along the plane of the chip, i.e., in a *in-plane* configuration (Fig. 1a). All the measured reflection scattering parameters have been normalized over the average value of the signal baseline measured off-resonance.

## Molecular spin samples

We use two samples of the molecular compounds shown in Fig. 1c. The first one is a diluted solid dispersion of $\alpha, \gamma$-bisdiphenylene-$\beta$-phenylallyl (BDPA, for short) organic radical into a polystyrene matrix, with a spin density of $\approx 1 \cdot 10^{15}$ spin/mm³. The sample was prepared as described in ref. 71 and then cut into a rectangular shape with a size $\approx 1.5 \times 1$ mm². The sample is placed on the resonator as shown in Fig. 1a. Each molecule has electronic spin $S = 1/2$, which

is due to its single unpaired electron[71,72]. Pure BDPA samples typically have antiferromagnetic exchange interaction occuring below 10 K[72–74], with a Curie-Weiss temperature between $T_C = -8$ K and $T_C = -1$ K, strongly dependent by the spin concentration and by the solvent or matrix used[72–74]. Although there are no reports for BDPA diluted in Polystyrene, due to the relatively high concentration, we can expect that a residual antiferromagnetic interaction still occurs among molecules. Therefore, our BDPA sample constitutes a prototypical TLS collection with essentially negligible magnetic anisotropy and no hyperfine splitting, which gives a single transition frequency[71,72].

The other sample is a single crystal of VOTPP with 2% concentration in its isostructural diamagnetic analog, TiO(TPP). Each molecule has an electronic spin $S = 1/2$ ground state and an additional hyperfine splitting due to the interaction with the $I = 7/2$ nuclear spin of the $^{51}$V ion (natural abundance: 99.75%). This results in a multiplet with eight $\{S_z, I_z\}$ electronuclear transitions, which can be exploited as eight independent spin ensembles. The magnetic properties and the electron spin resonance spectroscopy of this molecule have been previously reported in ref. 40. In particular, the hyperfine tensor shows uniaxial anisotropy with parallel component $A_\parallel = 477$ MHz $= 23$ mK and perpendicular component $A_\perp = 68$ MHz $= 8$ mK[40], respectively. These relatively large values, combined with the temperature of the experiments and the frequency of the resonator, give different thermal populations on the eight lines, thus decreasing the coupling rate $g_\mu$ for increasing line number $i$ (see Supplementary Information). The position of the VOTPP crystal (from #A to #D in Fig. 1a) is adjusted before each cooldown. Due to the experimental configuration used and the orientation of the molecules inside the unit cell, all molecules experience the same static magnetic field, which lies on the TPP plane and it is nearly perpendicular to the direction of the $V = 0$ double bound.

## Data availability

Experimental and theoretical data supporting our findings are available on Zenodo (https://doi.org/10.5281/zenodo.17609672)[75]. The molecular structures shown in Fig. 1b are available on the Cambridge Structural Database (CSD) by the Cambridge Chemical Structure Datacenter (CCSD), with deposition numbers 1310214[76] and 1847868[40], respectively.

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

## Acknowledgements

We thank Dr. Johan van Tol (National High Magnetic Field Laboratory, Florida, USA) for the preparation of BDPA sample. We thank Dr. Filippo Troiani (Institute of Nanoscience (NANO) of National Research Council (CNR)) for stimulating discussion. This work was funded by the National Quantum Science and Technology Institut (NQSTI) PE00000023 SPOKE 5- call N. 1 Project "Addressing molecular and Spins with MIcrowave puLsEs through Superconducting circuits for QUantum Information Processing (SMILE-SQUIP)" (M.A.), by the US Office of Naval Research award N62909-23-1-2079 project "Molecular Spin Quantum Technologies and Quantum Algorithms" (M.A.), and by the Army Research Office (ARO) (Grant No.

W911NF-19-1-0065) (S.S.). This work was also supported by the International Excellence Grants Program of KIT thorugh the University of Excellence concept (C.B.).

## Author contributions

The experiment was conceived and designed by C.B., A.G., and M.A. The theoretical framework has been developed by D.L., S.N., and S.S. The measurement were carried out by C.B. and S.G. at Karlsruhe Institute of Technology, under the supervision of W.W. The resonator were designed by D.R. Data fitting and simulation were done by D.L. and S.N. (with equal contributions). The VOTPP sample was prepared by F.S., who carried out also the EasySpin simulation of VOTPP. The manuscript was written by C.B. and D.L. with inputs from all coauthors. All authors contributed to the discussion of the results. The manuscript has been revised by all authors before submission.

## Competing interests

All authors declare no competing interests.
