## [Transparent Peer Review file · Nature Communications]

Observation of Perfect Absorption in Hyperfine Levels of Molecular Spins with Hermitian Subspaces

Corresponding Author: Dr Claudio Bonizzoni

Version 0:

Reviewer comments:

Reviewer #1

(Remarks to the Author)

This work investigated the perfect absorption phenomenon based on a non-Hermitian framework in a system consisting of spin-ensembles coupled to a microwave resonator. The manuscript presents the formalism of non-Hermitian Hamiltonian that describes the various sources of dissipation, and transformed it to a diagonal form under the polariton eigenbasis. The experiments demonstrated perfect absorption in both weak and strong coupling regimes, i.e., zero reflectance at the eigenfrequencies, for the cases of one and multiple spin ensembles. The resonance frequencies of spins can be controlled through a magnetic field while the resonator-spin coupling can be adjusted by position tuning. The conditions of perfect absorption were verified through simulation of dressed cavity feeding rates and dressed spin loss rates versus the applied magnetic field. Potential application of a microwave photon switch is discussed.

The perfect absorption has been demonstrated in a plethora of platforms. As mentioned in the introduction it has been explored in PT-symmetric systems, and here the manuscript demonstrated a PA in absence of PT symmetry. However, there is no clear connection between those two concepts, since the PA is a generalized critical coupling phenomenon that can be found by balancing the coupling loss with intrinsic loss.

While the theoretical framework built in this work clarifies how to find perfect absorption in a spin-resonator system, the demonstration is not essentially distinct from other previous works in other systems. The link of this work to PT symmetry is also not clear. Thus I do not recommend its publication in this form.

Detailed comments:

- 1) The demonstration of PA in the absence of PT-symmetry is natural, since PA is not related to PT-symmetry. The PT symmetry requires that whenever the zeros become real the poles are also real. Without PT symmetry, one can tailor the parameters to move zeros to the real axis which realizes PA, leaving the poles in the complex plane. Thus I do not see a clear necessity to highlight the connection between PA and PT symmetry.
- 2) The text before Eq. (4) assumes that the effective Hamiltonian is diagonalized under a polariton basis. However, since it involves a generic non-Hermitian description, we cannot neglect the possibility that the system reaches an exceptional point with degenerate eigenstates. In this situation, the system Hamiltonian given should take Jordan normal form instead. Then what would be the form for Eq. (4)?
- 3) Explain the PT symmetry condition mentioned in page 7, especially how this system satisfies PT symmetry.
- 4) What is the physical meaning of having a Hermitian subspace, and how is it related to previous works, especially the PA in the PT symmetry case, as authors claim it to be a special case of this work?
- 5) Regarding the application in microwave switch, though the modulation depth can reach 50dB, the on-state only leads to <0.2 transmission indicating a large loss. The limited bandwidth is another issue. These factors could hinder its practical use.

Reviewer #2

(Remarks to the Author)

This manuscript presents perfect absorption (PA) phenomena in passive open quantum systems based on molecular spins coupled to planar superconducting microwave resonators. The authors demonstrate that PA can be realized and understood in terms of Hermitian subspaces in the absence of parity-time (PT) symmetry. The theoretical framework is verified using two types of molecular spin systems: BDPA (a prototypical two-level system) and VOTPP (with multiple hyperfine transitions).

The results seem to be new in the subfield of molecular spintronics, but may not be new in more general field such as non-Hermitian physics. Therefore, I cannot recommend the publication in the present form unless the authors address my serious concerns as in the following:

(1) The theoretical framework including two non-hermitian hamiltonians ($H(NH)$ and $H(eff)$) which correspond respectively to poles and zeros of S_{11} is not original, as earlier work (Nat.Phys. 20, 1904 (2024)) has been adopted. In fact, Eq.(3) in this manuscript is same as Eq.(1) in Nat.Phys. 20, 1904 (2024) where the physical meaning of two effective non-Hermitian Hamiltonians are explained explicitly. The authors should cite earlier work and specify the difference if any.

(2) The authors observed PA and hence emphasize the importance of Hermitian subspaces in the absence of PT symmetry. However, this is rather obvious from previous works such as PRL 125, 147202 (2020) and Sci.Adv.9, eadg4730 (2023) using different terminology but essentially same underlying physics. And in the one-port configuration, this is sometimes termed as reflectionless scattering mode (RSM) which is a special case for complex zeros of S_{11} . See for example Phys. Rev. A 102, 063511 (2020).

(3) The authors claim in the discussion part that "previous findings in PT-symmetric systems (e.g. Ref. [18]) represent a particular case of the more general framework introduced here". I would say that "the more general framework" is already well-know in those previous works and they aimed at achieving the more strict and challenging PT-symmetric condition. I would recommend authors to rephrase so that readers will not be misled. Accordingly, I would encourage the authors to discuss whether the more strict PT-symmetry can be achieved in present work or not, and if not, explain the reason.

In addition, the format of present manuscript seems not to satisfy the requirement of Nat.Comm.

In summary, the theoretical framework of present manuscript seems to be not new and I think that the main advance in this paper lies in the experimental demonstration of these non-hermitian physics in the cryogenic platform with molecular spins which is indeed impressive.

Version 1:

Reviewer comments:

Reviewer #1

(Remarks to the Author)

The authors clarified the link between perfect absorption and PT symmetry under the non-Hermitian framework, and illustrated the Hermitian subspace by Fig. 5 with associated explanations. It is worth noting that PT symmetry discussed here is defined within the effective Hamiltonian that differs from the conventional one by the sign of the coupling term associated with the input port. The authors added the definition of the two Hamiltonians in the first paragraph in the introduction. To improve clarity further, it would be necessary to highlight the definition of PT symmetry in that paragraph since in most of previous works in non-Hermitian physics, PT symmetry was defined through the effective Hamiltonian of the system and described the behaviors of poles.

Though the theoretical framework is inherited from previous works, the authors claim that this work is the first example of perfect absorption demonstration in a quantum system made of spin centers coupled with microwave resonators. The demonstration in such new platforms could be interesting. Further, some potential applications in microwave photon control and detection have been discussed. Nevertheless, in the introduction the authors mentioned quantum information processing and sensing, but didn't explain how these applications are related to or motivate the engineering of PA in such a system. I would suggest further clarifying the motivation of this study in the introduction section.

In overall, the authors well addressed my questions related to physics and other issues along with appropriate revision. I would therefore recommend its publication, after making the necessary revision suggested above.

Reviewer #2

(Remarks to the Author)

The authors addressed most of my concerns and made revision to the manuscript. My additional suggestion is that it's better to specify which Hamiltonian (H_{res} or H_{rz}) authors refer to when they discuss the terms like PA, PT-symmetry, Hermitian subspace.

PA is only related to H_{rz} and has nothing to do with H_{res} ; but PT-symmetry is often referring to H_{res} in most of previous works and occasionally to H_{rz} (e.g., Ref.[27]). As a purely passive open system in present work, H_{res} must be PT-broken and hence it is trivial to say "in the absence of PT-symmetry for a passive system. Also, for the same reason, H_{res} has no any Hermitian subspace at all. Therefore, I recommend authors specify clearly that the term of "Hermitian subspace" applies only to H_{rz} , not generally to the system (which may mislead most of readers to mistakenly think H_{res} has any Hermitian subspace).

MANUSCRIPT: NCOMMS-25-43569A

TITLE: *Observation of Perfect Absorption in Hyperfine Levels of Molecular Spins with Hermitian Subspaces*

AUTHORS: Claudio Bonizzoni, Daniele Lamberto, Samuel Napoli, Simon Günzler, Dennis Rieger, Fabio Santanni, Alberto Ghirri, Wolfgang Wernsdorfer, Salvatore Savasta and Marco Affronte

AUTHORS' REPLY TO THE REVIEWERS

We thank the Editor for sending our manuscript out for review. We also thank Reviewers #1 and #2 for their careful evaluation of our manuscript and for their constructive and helpful comments.

In the following we first give a brief introduction to our reply and we then provide a point-by-point reply (black text) to all the comments raised by the Reviewers (blue text). A summary of all changes made is given at the end of the present rebuttal. Changes are marked with red color in the revised version of the manuscript.

*** INTRODUCTION TO OUR REPLY ***

The main concerns raised by both Reviewers are about the novelty of our proposed theoretical framework and the connections between the concepts of Perfect Absorption (PA), non-Hermitian systems and PT-symmetry. In the revised version we have done an extensive and careful effort in checking the suggested literature, in better clarifying their novelty and in the connections among the three above-mentioned concepts. In our opinion, this has brought to significant revisions and to a significant improvement of our manuscript.

We agree that PA can be studied in open quantum systems without the need to invoke non-Hermitian physics. Here we notice that in several works (see *e.g.*, [1–3] and *PRL 125, 147202 (2020)*, *Sci.Adv. 9, eadg4730 (2023)* suggested by Reviewer #2) scattering parameters and PA has been studied in the context of open systems with non-Hermitian physics. This motivates our interest in linking PA to non-Hermitian physics. In particular, we remark that we show that: i) the detuning can give the proper balance between radiative and non-radiative coupling rates for the observation of PA and, ii) this balance corresponds to an effective non-Hermitian Hamiltonian with an Hermitian subspace.

We agree with the Reviewers that our formalism based on effective Hamiltonians is not a new tool introduced by us. However, as detailed in our reply, we believe that there are several important differences between our findings and previous works. These are related to the experimental realization and the experimental conditions used (*e.g.*, YIG spheres in 3D cavities, temperature, absence of hyperfine interaction), to the observed coupling regime (*e.g.*, presence or not of polaritons) and in the type of interaction coupling spins and resonator (*e.g.*, dissipative coupling).

Next, we remark that our system does *not* satisfy the conditions required for PT-symmetry. To better highlight the connection between our results (in particular, the proposed Hermitian subspaces) and PT-symmetry we have prepared the plot in Fig. R1 of the present reply, which has also been added in the manuscript (new Figure 5). This additional simulation better clarifies how Hermitian subspaces can predict the occurrence of PT-symmetry conditions for a special choice of the parameters of our system.

Finally, we would like to further stress that we demonstrate PA on cryogenic platform based on microwave resonator coupled to molecular spins at milliKelvin temperature and in the single-photon regime. To the best of our knowledge, this not only corresponds to the first experimental demonstration achieved by using molecular spin centers but also on ensembles of paramagnetic centers with large set of spin degrees of freedom (electronic and nuclear spins). In the revised version we have further improved the discussion concerning the link between our experimental results and applications.

All considerations above are discussed in detail in the following.

*** REPLY TO REVIEWER #1 ***

This work investigated the perfect absorption phenomenon based on a non-Hermitian framework in a system consisting of spin-ensembles coupled to a microwave resonator. The manuscript presents the formalism of non-Hermitian Hamiltonian that describes the various sources of dissipation, and transformed it to a diagonal form under the polariton eigenbasis. The experiments demonstrated perfect absorption in both weak and strong coupling regimes, i.e., zero reflectance at the eigenfrequencies, for the cases of one and multiple spin ensembles. The resonance frequencies of spins can be controlled through a magnetic field while the resonator-spin coupling can be adjusted by position tuning. The conditions of perfect absorption were verified through simulation of dressed cavity feeding rates and dressed spin loss rates versus the applied magnetic field. Potential application of a microwave photon switch is discussed.

We thank Reviewer #1 for his/her thorough assessment of our work and for the constructive comments.

The perfect absorption has been demonstrated in a plethora of platforms. As mentioned in the introduction it has been explored in PT-symmetric systems, and here the manuscript demonstrated a PA in absence of PT symmetry. However, there is no clear connection between those two concepts, since the PA is a generalized critical coupling phenomenon that can be found by balancing the coupling loss with intrinsic loss.

We agree with Reviewer #1 that Perfect Absorption (PA) has been demonstrated in a plethora of platforms. However, as mentioned in the introduction of our reply, we emphasize that our work constitutes the first report of PA obtained using molecular spin centers and, more generally, also ensembles of paramagnetic centers with large set of spin degrees of freedom (including both electronic and nuclear spins).

We also agree that the PA condition can be analyzed in the context of linear scattering theory for open systems without the need to invoke non-Hermitian physics. However, open quantum systems are closely linked to non-Hermitian physics, and, in several works (see *e.g.*, [1–3]), PA has been widely studied in this context.

We explicitly highlight this connection in our work by showing that: i) the detuning can allow to balance the input port coupling rate with the intrinsic losses and, ii) this balance corresponds to an effective non-Hermitian Hamiltonian with an Hermitian subspace. This link has been achieved through the introduction of non-Hermitian effective Hamiltonians, which describe the zeros and the poles of the scattering matrix.

Here we remark that such formalism based on effective Hamiltonians is not a new tool introduced by us, but it is widely employed to link PA with non-Hermitian physics (see *e.g.*, Refs. [1–3]). In particular, the reflection coefficient has zero value when the frequency of the input signal coincides with one of the eigenvalues of the effective non-Hermitian Hamiltonian linked to the numerator of the S_{11} scattering parameter (former \hat{H}_{eff} , now \hat{H}_{RZ} , in Eq. (2) of the manuscript). This can happen only when the effective non-Hermitian Hamiltonian displays a real eigenvalue, such that it is possible to find a particular frequency of the input field that nullifies the numerator of S_{11} (see Eq. (3) of the manuscript). As a consequence, it turns out that many systems on which PA has been demonstrated can be described by non-Hermitian effective Hamiltonians displaying PT-symmetry. This can be achieved by tailoring the loss (and eventual gain) rates to satisfy the well-known PT-symmetry balance condition. The interest for such PT-symmetric Hamiltonians rises from the fact that, in the unbroken phase, they have real energy spectra, thus PA can be achieved when the frequency of the input field corresponds to one of these real eigenvalues.

We have add these considerations in the revised version of the manuscript to make them more explicit and clear.

While the theoretical framework built in this work clarifies how to find perfect absorption in a spin-resonator system, the demonstration is not essentially distinct from other previous works in other systems. The link of this work to PT symmetry is also not clear. Thus I do not recommend its publication in this form.

We have now significantly modified our manuscript in order to better clarify the link between our results and previous works, by including several new references and with additional text and discussions. We have also better clarified the novelty of our work with respect to literature and the link with PT-symmetry (see also previous comment and forthcoming ones).

Detailed Comment 1

The demonstration of PA in the absence of PT-symmetry is natural, since PA is not related to PT-symmetry. The PT symmetry requires that whenever the zeros become real the poles are also real. Without PT symmetry, one can tailor the parameters to move zeros to the real axis which realizes PA, leaving the poles in the complex plane. Thus I do not see a clear necessity to highlight the connection between PA and PT symmetry.

As already stated in response to previous comments, we concur with Reviewer #1 that the definition of PA is not inherently related to PT symmetry. However, we emphasize that numerous analogies with non-Hermitian physics, and in particular within PT-symmetric systems, have been identified in the recent literature. These parallels, though not implying a direct equivalence, provide a useful conceptual framework and the possibility to stimulate further theoretical and experimental research along this line. For these reasons, we consider it essential to draw comparisons with such recent developments. As mentioned in the points above, we have clarified this point in the revised manuscript accordingly.

Concerning the Reviewer's statement "*The PT symmetry requires that whenever the zeros become real the poles are also real*", we regret to have to respectfully disagree. Specifically, the structure of the reflection coefficient S_{11} induces the introduction of two distinct non-Hermitian Hamiltonians (denoted as \hat{H}_{NH} and \hat{H}_{eff} before revisions) and denoted now as \hat{H}_{res} and \hat{H}_{RZ} , whose eigenvalues correspond to the poles and zeros of the reflection coefficient, respectively. Although these two Hamiltonians share a similar form, they differ in the sign of the coupling term associated with the input port γ_r (as detailed in both the main text and the supplementary material). Consequently, it follows directly that the condition for real zeros (i.e., having real eigenvalues of \hat{H}_{RZ}) does not imply that the poles (i.e., eigenvalues of \hat{H}_{res}) are simultaneously real. This can be easily demonstrated in the case of a single spin coupled to a resonator mode by explicit calculation. Indeed, the corresponding non-Hermitian Hamiltonians (see Supplementary Eq. (S.6)) are:

$$H_{\text{RZ}} = \begin{pmatrix} \omega_0 - i\frac{-\gamma_r + \gamma_{\text{nr}}}{2} & g \\ g & \omega_s - i\frac{\gamma_s}{2} \end{pmatrix} \quad ; \quad H_{\text{res}} = \begin{pmatrix} \omega_0 - i\frac{\gamma_r + \gamma_{\text{nr}}}{2} & g \\ g & \omega_s - i\frac{\gamma_s}{2} \end{pmatrix}. \quad (\text{R.1})$$

Under PT-symmetry conditions for \hat{H}_{eff} , namely $\gamma_r - \gamma_{\text{nr}} = \gamma_s$ and $\omega_s = \omega_0$, these Hamiltonians reduces to

$$H_{\text{RZ}} = \begin{pmatrix} \omega_0 + i\frac{\gamma_s}{2} & g \\ g & \omega_0 - i\frac{\gamma_s}{2} \end{pmatrix} \quad ; \quad H_{\text{res}} = \begin{pmatrix} \omega_0 + i\frac{\gamma_s - 2\gamma_r}{2} & g \\ g & \omega_0 - i\frac{\gamma_s}{2} \end{pmatrix}. \quad (\text{R.2})$$

The corresponding eigenfrequencies are

$$H_{\text{RZ}} \implies \tilde{\Omega} = \omega_0 \pm \sqrt{g^2 - \frac{\gamma^2}{4}}, \quad (\text{R.3})$$

$$H_{\text{res}} \implies \Omega = \omega_0 - i\frac{\gamma_r}{2} \pm \sqrt{g^2 - \left(\frac{\gamma_s - \gamma_r}{2}\right)^2}. \quad (\text{R.4})$$

These expressions clearly show that the zeros and poles of the reflection coefficient cannot be simultaneously real, even under PT-symmetric conditions.

Moreover, in the absence of exact PT symmetry, one can utilize frequency detuning to render one of the two complex eigenvalues real (*Hermitian subspace*), thereby enabling the realization of PA. This mechanism, which does not rely on PT symmetry, broadens the range of physical systems where PA-like behavior can be observed with respect to PT-symmetric systems.

Detailed Comment 2

The text before Eq. (4) assumes that the effective Hamiltonian is diagonalized under a polariton basis. However, since it involves a generic non-Hermitian description, we cannot neglect the possibility that the system reaches an exceptional point with degenerate eigenstates. In this situation, the system Hamiltonian given should take Jordan normal form instead. Then what would be the form for Eq. (4)?

As correctly noted by the Reviewer, the derivation of Eq. (4) relies on the assumption that the Hamiltonian can be approximately diagonalized in the polariton basis, at least within the strong coupling regime. However, despite the non-Hermitian nature of the Hamiltonian, it does not display PT symmetry, since the condition $\gamma_r - \gamma_{nr} = \gamma_s$ is not satisfied. Consequently, the system cannot exhibit an exceptional point. To clarify this further we notice that the occurrence of an Exceptional Point requires the coalescence of *both* eigenvalues and eigenstates. However, as discussed at several points in the manuscript, within a generic Hermitian subspace, only one eigenvalue is real while the other remains complex. This precludes the coalescence of the eigenvalues, and thus rules out the formation of an Exceptional Point. This behavior is explicitly illustrated in the single-spin case shown in Fig. 2(d), which plots the system’s dressed decay rates (corresponding to the imaginary parts of the eigenvalues) introduced in Eq. (5), further supporting the absence of such a degeneracy.

In conclusion, the conditions necessary for the emergence of a Jordan block are not fulfilled in our system.

Detailed Comment 3

Explain the PT symmetry condition mentioned in page 7, especially how this system satisfies PT symmetry.

We would like to emphasize again that our system does *not* satisfy the conditions required for PT-symmetry. For a system comprising two coupled bosonic modes, such as a resonator mode and a collective spin excitation in the low-excitation regime, PT symmetry of the effective non-Hermitian Hamiltonian requires that $\omega_0 = \omega_s$ and $\gamma_r - \gamma_{nr} = \gamma_s$. These conditions ensure that the effective Hamiltonian is PT-symmetric (see also the reply to “Detailed Comment 1”). In contrast, our system does not fulfill these conditions. In particular, we can tune the resonance frequency of spins and hence the detuning, but our system displays spin decay rates larger than the resonator radiative decay rate.

FIG. R1. Surface plot of the imaginary parts of the complex eigenfrequencies, $\Im(\tilde{\Omega})$, as functions of the detuning Δ and the spin decay rate γ_s . The red lines indicate the intersections of surfaces with the plane $\Im(\tilde{\Omega}) = 0$, corresponding to the emergence of Hermitian subspaces. PT symmetry is realized only at the specific point in parameter space where $\gamma_s = \gamma_r - \gamma_{nr}$, marked by the red triangle. The parameters γ_r , γ_{nr} and g are fixed to the experimentally determined values reported in Table S1 of the Supplementary Materials.

To better address this point we have prepared the surface surface plot in Fig. R1, which illustrates the behavior of the imaginary parts of the two complex eigenfrequencies, $\Im(\tilde{\Omega})$, as functions of the detuning Δ and the spin decay rate γ_s , with γ_r , γ_{nr} and the coupling strength, g , held fixed at the experimentally determined values (see Table S1 in the Supplementary Information). The red curves indicate the intersection of these surfaces with the $\Im(\tilde{\Omega}) = 0$ plane (blue), corresponding to parameter sets for which a Hermitian subspace is realized (i.e., one of the eigenfrequencies is purely real). As discussed in the replies to previous comments, whenever one eigenvalue becomes real, the other retains a non-zero imaginary part, and hence the system does not exhibit full Hermiticity. The simultaneous coalescence of

the two Hermitian subspaces (the two red curves associated to the two different polariton branches) occurs exclusively at the point where both $\gamma_s = \gamma_r - \gamma_{nr}$ and $\Delta = 0$, which precisely defines the conditions for PT symmetry. This point is highlighted by the red triangle in Fig. R1. We have added the considerations above and Fig. R1 to the revised version (new Figure 5).

Detailed Comment 4

What is the physical meaning of having a Hermitian subspace, and how is it related to previous works, especially the PA in the PT symmetry case, as authors claim it to be a special case of this work?

Hermitian subspaces are realized for regions of parameter space in which at least one of the complex eigenfrequencies of a non-Hermitian Hamiltonian becomes purely real, i.e., its imaginary part vanishes.

The connection between Hermitian subspaces and PA is then straightforward: when a Hermitian subspace is present, it is possible to tune the frequency of the input probing field to match the real eigenvalue which corresponds to the Hermitian subspace. Under this condition, the reflection coefficient S_{11} can vanish, signaling the occurrence of PA. Conversely, if PA is observed in a system described by a non-Hermitian Hamiltonian (not necessarily PT-symmetric), then a Hermitian subspace must be present. This is because the condition $S_{11} = 0$ requires at least one eigenfrequency of the effective Hamiltonian to be real (see Eq. (3)), which by definition implies the existence of a Hermitian subspace associated with that eigenvalue. In summary, PA occurs if and only if a Hermitian subspace is present.

We now turn to the relationship between Hermitian subspaces and PT symmetry, and clarify why PT-symmetry can be regarded as a particular case within the broader framework of Hermitian subspaces. First, it is important to note that PT-symmetry, in the context of this manuscript, is well-defined only for systems composed of two coupled bosonic modes, such as a cavity mode and a spin ensemble in the low-excitation regime. In contrast, the notion of Hermitian subspaces is more general and can be naturally extended to systems with arbitrary numbers of interacting subsystems, as it is based solely on the condition that the imaginary part of at least one eigenfrequency vanishes. This distinction is crucial for interpreting the experimental data from the VOTPP sample discussed in the manuscript, where the system involves more than two interacting components (1 resonator mode and 8 spin ensembles), and thus PT-symmetry balance condition is no longer applicable. In this case, the concept of Hermitian subspaces offers a useful and intuitive framework to describe the observed phenomena and to correctly predict the occurrence of PA.

As shown in the reply to the Detailed comment 3, even in the case of a two-mode system Hermitian subspaces provide a generalization of PT-symmetry. This is also illustrated in Fig. R1, which shows the imaginary parts of the complex eigenfrequencies $\Im(\tilde{\Omega})$ as functions of the detuning Δ and the spin decay rate γ_s . Hermitian subspaces correspond to the parameter curves along which one eigenvalue becomes purely real (red lines). In contrast, PT-symmetry is realized only at a single point in this parameter space, specifically, when $\gamma_s = \gamma_r - \gamma_{nr}$ and $\Delta = 0$ (marked by the red triangle in Fig. R1).

This analysis highlights the practical advantages of the Hermitian subspace framework over PT-symmetry. While PT-symmetry ensures that both eigenvalues are real (i.e., both dips in the reflection spectrum reach zero), it imposes stringent conditions on the decay rates, which are often difficult to achieve in realistic experimental setups. On the other hand, the realization of Hermitian subspaces requires only that a single eigenvalue becomes real (and thus only the corresponding dip reaches zero), which is sufficient for practical implementations of PA-based devices, such as switches and modulators. These considerations have been added to the revised version of the manuscript.

Detailed Comment 5

Regarding the application in microwave switch, though the modulation depth can reach 50dB, the on-state only leads to <0.2 transmission indicating a large loss. The limited bandwidth is another issue. These factors could hinder its practical use.

We first notice that only reflection measurements can be done on our system, since we use only one-port device (see Fig. 1 of main text), therefore we have no transmission data.

The potential application of our results to a microwave switch is show in Fig. 6 (former Fig. 5 before revision) of the main paper. When the system is in the strong coupling regime (Fig.2 and Fig.4a,b,g), the reflection can be switched from ~ 1 to exactly zero (due to PA). The modulation depth ~ 50 dB correspond to this result (Fig. 6a,b).

In the weak coupling regime (Fig.4d,h,i) PA is reached only for the first four lines of VOTPP ($B_0 < 0.345$ T), where zero reflection is still clearly visible. In the previous version of the manuscript we reported in Fig.5d (now Fig. 6) an example of reflection without PA, where the minimum value 0.2 mentioned by the Reviewer was shown. Therefore, "*the on-state only leads to <0.2 transmission indicating a large loss*" mentioned by the Reviewer (former Fig.5d) did not correspond to PA. While we agree with the reviewer this corresponds to large losses, we would like to clarify that the application to microwave single photon switches relies in the use of our hybrid device when the PA condition is met, so not in the case shown in former Fig.5d. To avoid possible confusion, we have rephrased the caption of Fig. 6 (former Fig. 5 before revisions), removed panels c and d from it and changed the corresponding text accordingly.

Concerning the bandwidth, the Reviewer is probably referring to the width (Full width at Half Maximum or Half Width at Half Maximum) of the dips under PA condition in Fig. 6.a,b,c (former Fig. 5). We agree with Reviewer #1 that the working bandwidth could be a limiting factor for practical applications, especially when a single spin ensemble is used (as in Fig. 2). However, our system and PA condition relies on the interplay of several different physical parameters, which can be independently tuned. This means that the bandwidth can be slightly tuned depending on the needed value, giving a certain flexibility. In addition, the use of multiple spin (sub) ensembles with large nuclear spin values (as in Fig. 4) gives a large set of possible working points in the same system both in the coherent coupling as well as in the weak coupling regime.

We finally notice that using our system as a switch would require operation in time domain, *e.g.* with an additional local modulation field. In this case the available working bandwidth depends by the repetition time of a working cycle from the ON to OFF state or viceversa. This is mainly imposed by the way the local field is applied and switched on the system (coils, microcoils, DC bias lines) rather than the system itself.

*** REPLY TO REVIEWER #2 ***

This manuscript presents perfect absorption (PA) phenomena in passive open quantum systems based on molecular spins coupled to planar superconducting microwave resonators. The authors demonstrate that PA can be realized and understood in terms of Hermitian subspaces in the absence of parity-time (PT) symmetry. The theoretical framework is verified using two types of molecular spin systems: BDPA (a prototypical two-level system) and VOTPP (with multiple hyperfine transitions).

We thank Reviewer #2 for the accurate description of our work.

The results seem to be new in the subfield of molecular spintronics, but may not be new in more general field such as non-Hermitian physics. Therefore, I cannot recommend the publication in the present form unless the authors address my serious concerns as in the following

We thank Reviewer #2 for appreciating the novelty of our results in the context of molecular spintronics. We also thank the Reviewer for his/her very useful report, which strongly helped us to better frame our work in the context of the existing literature.

Comment 1

*The theoretical framework including two non-hermitian hamiltonians ($H(NH)$ and $H(\text{eff})$) which correspond respectively to poles and zeros of S_{11} is not original, as earlier work (*Nat.Phys.* 20, 1904 (2024)) has been adopted. In fact, Eq.(3) in this manuscript is same as Eq.(1) in *Nat.Phys.* 20, 1904 (2024) where the physical meaning of two effective non-Hermitian Hamiltonians are explained explicitly. The authors should cite earlier work and specify the difference if any.*

We thank Reviewer #2 for bringing to our attention the recent work *Nat. Phys.* 20, 1904 (2024), which we have now included as a relevant reference in the revised manuscript. In the light of this mentioned work we acknowledge that our theoretical framework is not fully original and bears strong similarities to the one developed in that study. We have now properly pointed out this connection in the revised version and changed some nomenclature accordingly (see also our reply to Comment 2).

However, we would also like to highlight several important distinctions with respect to our work, mainly concerning the investigated system and the results. From an experimental standpoint, as correctly noted by the Reviewer, our system consists of molecular spin centers coherently coupled to a planar superconducting microwave resonator. The system operates at milliKelvin temperature and in the single microwave photon regime. In contrast, the system studied in *Nat. Phys.* 20, 1904 (2024) comprises two magnon modes, realized in YIG spheres, which are coupled via their dissipative interaction with a common waveguide. This fundamental difference in the experimental setup also has significant implications on the theoretical side, as it leads to qualitatively distinct coupling mechanisms in the respective models. In particular, the coupling in *Nat. Phys.* 20, 1904 (2024) is dissipative in nature, arising from the indirect interaction between the two YIG spheres mediated by the common waveguide. In contrast, our system features a coherent coupling between the spin ensemble and the resonator mode, directly implemented via the spatial overlap of the resonator field mode and the molecular spins. As a result, the structure of the effective non-Hermitian Hamiltonians in the two works differs significantly, as it can be verified by a direct confrontation between \hat{H}_{eff} (now also \hat{H}_{RZ} in order to align with the existing literature) of our work in the case of two modes only (Eq. (R.1)) and \hat{H}_{RZ} of *Nat. Phys.* 20, 1904 (2024) (Eq. (5)). This difference becomes particularly evident under PT-symmetric conditions, where the terms responsible for enforcing symmetry balance differ between the two models (see, for example, the comparison between Eqs. (R.2) and (R.3) in our reply, and Eq. (6) in *Nat. Phys.* 20, 1904 (2024)). Therefore, although both approaches fall within the broader framework of non-Hermitian physics, the underlying coupling mechanisms and the resulting phenomenology are fundamentally different. These differences offer complementary perspectives on perfect absorption and PT-like phenomena. In this regard, our work and *Nat. Phys.* 20, 1904 (2024), together with related studies (such as *Nat. Commun.* 8, 1368 (2017), already cited in the manuscript, which also bears some similarities in the Hamiltonian description), contribute to expand the field of PA-related non-Hermitian physics and build a more comprehensive understanding, also across different physical platforms. The revised version of the manuscript includes a brief comparison between our results and these mentioned works.

We would like to thank the Reviewer for his/her valuable suggestion (see also the forthcoming replies), which have helped us expand the discussion of relevant literature and more clearly position our work within the broader context of recent developments in the field. We believe that this has significantly improved the framing and relevance of our manuscript.

Comment 2

The authors observed PA and hence emphasize the importance of Hermitian subspaces in the absence of PT symmetry. However, this is rather obvious from previous works such as PRL 125, 147202 (2020) and Sci.Adv.9, eadg4730 (2023) using different terminology but essentially same underlying physics. And in the one-port configuration, this is sometimes termed as reflectionless scattering mode (RSM) which is a special case for complex zeros of S11. See for example Phys. Rev. A 102, 063511 (2020).

These two additional papers suggested by Reviewer #2 (*PRL 125, 147202 (2020)* and *Sci.Adv.9, eadg4730 (2023)*) represent very interesting examples of how peculiar scattering properties of open systems can be described and interpreted in terms of the physics of non-Hermitian systems. We thank the Reviewer for suggesting these references which have been added to our references. Although these two papers are related to our work, we observe that they study physical processes and systems that differs significantly from ours (see also authors' reply to the previous comment). In the first paper, the coupling between the two components (the cavity and the YIG sphere) is purely dissipative. In contrast, in our work there is no dissipative coupling between the two components: the coupling is purely coherent. As a result, *PRL 125, 147202 (2020)* does not evidence any avoided level crossing, in contrast to our polaritons. *Sci.Adv.9, eadg4730 (2023)* studies a more complex system with three components (two YIG spheres and an electromagnetic resonator), with one of the YIG spheres embedded in the continuum of an open waveguide. Also in this case, dissipative coupling play a relevant role, although coherent coupling is also present. Most of the results do not display any avoided level crossing in the real part of the system eigenvalues, in contrast to our main results. Both papers describe components strongly interacting with open waveguides, so that their properties can be described in terms of bound states in the continuum (BIC). In our work, we mainly study strongly and coherently interacting components giving rise to avoided level crossings (observable in the spectra) with a component weakly coupled to the external continuum. So, our system can better be described as an open quantum polariton system.

Concerning the one-port configuration and the reflectionless scattering mode (RSM), we agree with Reviewer #2 that PA in the absence of PT-symmetry is not a new concept. For example, Ref. [4], which was already cited in our work, has theoretically shown how to reach PA in the absence of PT-symmetry in interacting light-matter systems, although not explicitly linking it with non-Hermitian physics. The paper *Phys. Rev. A 102, 063511 (2020)* suggested by Reviewer #2 provides a general theory for RSM modes and shows how these modes are not necessarily related to PT-symmetry. More generally, the occurrence of these scattering modes can be studied even without introducing the framework of non-Hermitian physics. The choice of introducing the concept of Hermitian subspaces origins from the fact that we wish to better frame our work in a non-Hermitian physics framework: open quantum systems are closely linked to non-Hermitian physics, and, in several works (especially the most recent ones, included those suggested by Reviewer #2), PA has been widely studied in this context.

In conclusion, that is new in our manuscript, to the best of our knowledge is: i) the quantum system based on molecular spins where PA has been observed; ii) the observation of PA in the absence of PT symmetry in a purely polariton system based on the coherent coupling between light and matter excitations; iii) the analysis showing that in these systems, PA in the absence of the impedance matching condition involving the bare damping rates (that is $\gamma_r = \gamma_{nr} + \gamma_s$, corresponding to PT symmetry), can be achieved controlling the photon and spin content of a polariton mode (see also authors' reply to Comment 3).

Comment 3

The authors claim in the discussion part that "previous findings in PT-symmetric systems (e.g. Ref. [18]) represent a particular case of the more general framework introduced here". I would say that "the more general framework" is already well-know in those previous works and they aimed at achieving the more strict and challenging PT-symmetric condition. I would recommend authors to rephrase so that readers will not be misled. Accordingly, I would encourage the authors to discuss whether the more strict PT-symmetry can be achieved in present work or not, and if not,

explain the reason.

We thank Reviewer #2 for pointing out that the cited phrase in our manuscript could be potentially misleading. In line with their suggestion, we have revised the text to improve clarity and avoid ambiguity.

As the Reviewer correctly stated, and as we have also emphasized in the points above, PT-symmetry constitutes a special case within the broader framework of effective non-Hermitian Hamiltonians, whose eigenvalues correspond to the zeros of the S_{11} reflection coefficient. However, due to the stringent and often experimentally challenging requirements needed to realize PT-symmetry we believe that the ability to employ detuning to compensate for the lack of balancing between the decay (or eventual gain) rates constitutes a significant and distinctive feature of our study. Moreover, the use of the dressed loss rates (Eq. (5) in the manuscript) provides a novel and physically transparent interpretation. To the best of our knowledge, this concept has not been introduced previously and can serve as a further link to the PT-symmetry condition, which can itself be viewed as a particular instance of a more general “dressed” balance between decay and gain rates (see also Fig. 2(d) of the manuscript and the new Fig. 5 of the manuscript, Fig. R1 of the represent rebuttal).

Finally we wish to clarify that “real” PT symmetry can, in principle, be achieved in our system based on the BDPA sample by carefully tuning both the coupling strength (via the position of the sample on the resonator) and the balance of radiative and non-radiative decay rates (influenced by the position of the antenna relative to the resonator). Nonetheless, achieving such precise balancing can be challenging. Furthermore, in more complex systems involving more than two interacting subsystems, such as the VOTPP sample discussed in our manuscript, the very notion of PT symmetry becomes ill-defined. This further motivates the need for a broader conceptual framework, such as the one discussed, which cannot be reduced to conventional PT symmetry alone.

Additional Comment

In addition, the format of present manuscript seems not to satisfy the requirement of Nat.Comm.

In summary, the theoretical framework of present manuscript seems to be not new and I think that the main advance in this paper lies in the experimental demonstration of these non-Hermitian physics in the cryogenic platform with molecular spins which is indeed impressive.

According to the website of Nature Communications, no specific format requirements were needed at our initial and first submission stage (see <https://www.nature.com/ncomms/submit/article>). We have checked before submission that the manuscript did not exceed the maximum number of suggested words for text and methods, as well as the maximum number of visual elements (figures and tables) and references. During revision, following the suggestions of the Reviewers we have included additional portions of text, one more figure and new references. To better meet the guidelines we have now moved the former Section 1 ("Theoretical modeling") into Section 2 ("Results") as a subsection with new title "Section 2.1.Theoretical modeling". A new Section 1 (Introduction) was added.

We would like to thank the Reviewer for considering our experimental realization impressive.

*** LIST OF CHANGES ***

Nomenclature

- Change of the name of the threshold value of the coupling constant for which the crossing in the imaginary parts disappears from g_{EP} to g_{th} , in order to avoid confusion. The nomenclature g_{EP} has been used only in considerations regarding PT symmetry.
- Change of \hat{H}_{NH} to \hat{H}_{res} and \hat{H}_{eff} to \hat{H}_{RZ} to align with recent literature.

Main Text

- **Abstract:** The abstract has been revised in order to align with the changes in the manuscript.
- **Introduction:** We have modified the introduction to include the literature suggested by the Reviewers and to better highlight further research on molecular spins.
- **Section 1. Theoretical modeling:** This section has now become a subsection of "Results" section, with new title "Section 2.1 Theoretical modeling". All sections have been renumbered consequently.
- **Figure 1:** Change of Fig. 1(d) with a zoomed version to better highlight the energy levels of interest. Label in Fig. 1(a) has been changed from "BDPA" to "SPINS" for better clarity.
- **Figure 5:** This is a new Figure added to better highlight the connection between our results and the PT-Symmetry, according to the comments raised by the Reviewers and our reply.
- **Figure 6 (former Figure 5, before revisions):** Following Comment 5 by Reviewer #1, we have removed panels c and d to avoid possible confusion for the reader.
- **Discussion:** we have modified the discussion by adding a paragraph which explicitly highlight the connection between our results and PT-symmetry. We have now improved and expanded our discussion on the applications of our experimental results.

-
- [1] D. Zhang, X.-Q. Luo, Y.-P. Wang, T.-F. Li, and J. Q. You, Observation of the exceptional point in cavity magnon-polaritons, *Nature Communications* **8**, 1368 (2017).
- [2] W. R. Sweeney, C. W. Hsu, and A. D. Stone, Theory of reflectionless scattering modes, *Phys. Rev. A* **102**, 063511 (2020).
- [3] Z. Rao, C. Meng, Y. Han, L. Zhu, K. Ding, and Z. An, Braiding reflectionless states in non-hermitian magnonics, *Nature Physics* **20**, 1904 (2024).
- [4] S. Zanotto and A. Tredicucci, Universal lineshapes at the crossover between weak and strong critical coupling in fano-resonant coupled oscillators, *Scientific Reports* **6**, 24592 (2016).

MANUSCRIPT: NCOMMS-25-43569A

TITLE: *Observation of Perfect Absorption in Hyperfine Levels of Molecular Spins with Hermitian Subspaces*

AUTHORS: Claudio Bonizzoni, Daniele Lamberto, Samuel Napoli, Simon Günzler, Dennis Rieger, Fabio Santanni, Alberto Ghirri, Wolfgang Wernsdorfer, Salvatore Savasta and Marco Affronte

AUTHORS' REPLY TO THE REVIEWERS

We thank the Editor for overseeing the review process. We thank again Reviewers #1 and #2 for their additional comments and valuable suggestions.

In the following we provide a point-by-point reply (black text) to all the suggestions and comments raised by the Reviewers (blue text). A summary of all changes made is given at the end of the present document. Changes are marked with red color in the revised manuscript (pdf version with changes marked).

*** REPLY TO REVIEWER #1 ***

The authors clarified the link between perfect absorption and PT symmetry under the non-Hermitian framework, and illustrated the Hermitian subspace by Fig. 5 with associated explanations. It is worth noting that PT symmetry discussed here is defined within the effective Hamiltonian that differs from the conventional one by the sign of the coupling term associated with the input port. The authors added the definition of the two Hamiltonians in the first paragraph in the introduction. To improve clarity further, it would be necessary to highlight the definition of PT symmetry in that paragraph since in most of previous works in non-Hermitian physics, PT symmetry was defined through the effective Hamiltonian of the system and described the behaviors of poles.

We agree with Reviewer #1 that the PT-symmetry discussed in our work pertains to the effective Hamiltonian \hat{H}_{RZ} , in which the sign of the coupling associated with the input port is reversed. Following his/her suggestion, we have now clarified in the Introduction how PT-symmetry emerges from \hat{H}_{RZ} and also its connection to the effective gain-loss balance, since our system is entirely passive and thus this symmetry cannot be associated with the behavior of the poles.

Though the theoretical framework is inherited from previous works, the authors claim that this work is the first example of perfect absorption demonstration in a quantum system made of spin centers coupled with microwave resonators. The demonstration in such new platforms could be interesting. Further, some potential applications in microwave photon control and detection have been discussed. Nevertheless, in the introduction the authors mentioned quantum information processing and sensing, but didn't explain how these applications are related to or motivate the engineering of PA in such a system. I would suggest further clarifying the motivation of this study in the introduction section.

The original intention of the Introduction was to introduce molecular spin centers to the reader and to give examples of literature in which these spins systems have been already demonstrated to hold potential for applications. This includes also quantum sensing and quantum information processing (with also works from some author of the present manuscript). In the Discussion section, following the previous suggestion of Reviewers, we already commented on the possible implementation of sensing schemes (slope detection, Dark Matter Axion search) or of single photons switches and modulators involving Perfect Absorption (PA).

Concerning possible motivations and interest related to quantum sensing and quantum information processing we first notice that working with molecular spin qubits and, more generally, with solid state spin qubits necessary requires to take into account the open nature of the system and to find strategies to efficiently tailoring the photon-spin transduction (i.e., the way photon/s excitation/s are transferred and retrieved from the qubits) . Interactions with the environment, including *e.g.* relaxation channel/s and control channels to initialize, drive or readout the system, should be properly taken into account. For instance, with our model is it still possible to properly include and predict

the effect of all radiative and non-radiative dissipation channels and of the effective gain given by the microwave antenna even when multiple sub-ensembles are considered. This can already stimulate further studies and the design of experiments for quantum sensing or for quantum information processing.

It is worth noticing that quantum sensing and quantum information processing would require to bring the system out from its steady state with pulsed-wave excitations (microwave pulse sequences in our experimental implementation) and to investigate it in time-domain. This additional investigations were not reported in our manuscript and are beyond the scope of the present work. This also explains why no applications to these topics were discussed in the previous version. Nevertheless, we can foresee a potential interest by considering that the capability of planar superconducting microwave resonators to address spins with pulses have been already demonstrated with several spin centers (including molecular ones, see Ref. [50-58] of the present version) and over a wide range of experimental conditions, from room temperature down to cryogenic ones. The PA of single microwave photons reported in our work corresponds to the case in which the incoming photon is fully transferred to non-radiative channels: this could be exploited to perform the reset of the spins of the system or to decouple them from radiative losses, under approaches similar to what reported in [Zhang *et al. Nat. Comm.* 6, 8914 (2015); Bienfait *et al. Nature* 531, 74 (2016); Putz *et al. Nat. Phys.* 10, 720 (2014)]. At the same time we stress that, due to the dissipative nature of the non-radiative channels (in the way these are experimentally implemented in our manuscript), designing suitable pulse sequences for exploiting spin coherence in proximity or at the PA point, might result in the lost of coherence. As mentioned above this is would of course deserve an additional and dedicated work, which is clearly beyond the scope of our manuscript.

Due to the considerations above and following the suggestion of the Reviewer, we have now added a short comment in the Introduction about the potential interest of our results for quantum sensing and quantum information processing.

In overall, the authors well addressed my questions related to physics and other issues along with appropriate revision. I would therefore recommend its publication, after making the necessary revision suggested above.

We sincerely thank the Reviewer for his/her positive assessment of our work and for the constructive feedback provided. We have carefully implemented all the suggested revisions as requested.

***** REPLY TO REVIEWER #2 *****

The authors addressed most of my concerns and made revision to the manuscript. My additional suggestion is that it's better to specify which Hamiltonian (Hres or Hrz) authors refer to when they discuss the terms like PA, PT-symmetry, Hermitian subspace. PA is only related to Hrz and has nothing to do with Hres; but PT-symmetry is often referring to Hres in most of previous works and occasionally to Hrz (e.g., Ref.[27]). As a purely passive open system in present work, Hres must be PT-broken and hence it is trivial to say "in the absence of PT-symmetry for a passive system. Also, for the same reason, Hres has no any Hermitian subspace at all. Therefore, I recommend authors specify clearly that the term of "Hermitian subspace" applies only to Hrz, not generally to the system (which may mislead most of readers to mistakenly think Hres has any Hermitian subspace).

We thank the Reviewer for his/her valuable suggestion, which has helped us improve the clarity of the manuscript. We have introduced additional clarifications throughout the text to specify that references to PT-symmetry and Hermitian subspaces pertain to \hat{H}_{RZ} . As correctly noted by the Reviewer, some of these explanations may appear redundant to a specialized reader. However, we chose to retain them to make the discussion more accessible to a broader audience.

***** LIST OF CHANGES *******Main Text**

- **Introduction, Section 2.2 (The Single Ensemble Case) and Discussion:** We have revised and expanded the Introduction to more clearly introduce the effective Hamiltonians \hat{H}_{res} and \hat{H}_{RZ} , and to briefly introduce the role of PT-symmetry in relation to \hat{H}_{RZ} . In addition, brief clarifications have been added throughout the manuscript, where appropriate, to further emphasize this connection between PT-symmetry and \hat{H}_{RZ} .
 - **Introduction** We have added a brief comment about the potential interest for quantum sensing and for quantum information processing, according to our reply to Reviewer #1.
 - **Text and Captions** We edited the calls to Supplementary Tables according to the guidelines in the authors checklist (i.e. as Supplementary Figure xx).
 - **References** We edited the references to comply with the guidelines in the authors' checklist.
 - **Section order** Section order has been rearranged to comply with the authors' checklist and by properly including the necessary declarations and statements.
-